# Acceleration Exists! Optimization Problems When Oracle Can Only Compare Objective Function Values

**Aleksandr Lobanov**
MIPT, Skoltech, ISP RAS
lobbsasha@mail.ru

**Alexander Gasnikov**
Innopolis, MIPT, MI RAS
gasnikov@yandex.ru

**Andrei Krasnov**
MIPT, Innopolis
krasnov.an@phystech.edu

## Abstract

Frequently, the burgeoning field of black-box optimization encounters challenges due to a limited understanding of the mechanisms of the objective function. To address such problems, in this work we focus on the deterministic concept of Order Oracle, which only utilizes order access between function values (possibly with some bounded noise), but without assuming access to their values. As theoretical results, we propose a new approach to create non-accelerated optimization algorithms (obtained by integrating Order Oracle into existing optimization "tools") in non-convex, convex, and strongly convex settings that are as good as both SOTA coordinate algorithms with first-order oracle and SOTA algorithms with Order Oracle up to logarithm factor. Moreover, using the proposed approach, *we provide the first accelerated optimization algorithm using the Order Oracle*. And also, using an already different approach we provide the asymptotic convergence of *the first algorithm with the stochastic Order Oracle concept*. Finally, our theoretical results demonstrate effectiveness of proposed algorithms through numerical experiments.

## 1 Introduction

The black box problem has garnered extensive attention in diverse scientific and engineering domains, reflecting the challenge of optimizing systems with complex and opaque objective functions, prompting the exploration of innovative solutions Conn et al. (2009); Kimiaei and Neumaier (2022).

This paper focuses on solving a standard general optimization problem in the following form:

$$\min_{x \in \mathbb{R}^d} \left\{ f(x) := \mathbb{E}_{\xi \sim \mathcal{D}} f_\xi(x) \right\}, \tag{1}$$

where $f : \mathbb{R}^d \to \mathbb{R}$ is a possibly non-convex, possibly stochastic function. This problem configuration encompasses a broad range of applications in ML scenarios, e.g. empirical risk minimization, where $\mathcal{D}$ denotes distribution across training data points, and $f_\xi(x)$ represents loss of model $x$ on data point $\xi$.

When the objective function $f(x)$ has exclusive access to a zero-order oracle Rosenbrock (1960), problem (1) falls under the classification of a black-box optimization problem. This class of problems is actively studied in various application settings, e.g., deep learning Chen et al. (2017); Gao et al. (2018), federated learning Dai et al. (2020); Alashqar et al. (2023); Patel et al. (2022), reinforcement learning Choromanski et al. (2018); Mania et al. (2018), overparameterized models Lobanov and Gasnikov (2023), online optimization (Agarwal et al., 2010; Bach and Perchet, 2016; Akhavan et al., 2022), multi-armed bandits Shamir (2017); Lattimore and Gyorgy (2021), hyperparameter settings Bergstra and Bengio (2012); Hernández-Lobato et al. (2014); Nguyen and Balasubramanian (2022), and control system performance optimization Bansal et al. (2017); Xu et al. (2022). It is important to highlight that the zero-order oracle presupposes awareness of the objective function's value $f(x_k)$ at a specific point $x_k$ (which may be inexact), enabling the development of effective gradient-free algorithms tailored to various problem scenarios Gasnikov et al. (2022a).

Table 1: Comparison of oracle complexity for the methods proposed in this work with SOTA methods both in the coordinate descent class and in the Order Oracle concept (2). Notation: $F_0 = f(x_0) - f(x^*)$; $R = \|x_0 - x^*\|$; $R_{[1-\alpha]} = R_{1-\alpha}(x_0) = \sup_{x \in \mathbb{R}^d : f(x) \leq f(x_0)} \|x - x^*\|_{[1-\alpha]}$; $\varepsilon =$ desired accuracy of problem solving.

| Reference | Nesterov (2012) | Gorbunov et al. (2019) | Saha et al. (2021) | Tang et al. (2023) | This paper | |
|---|---|---|---|---|---|---|
| Non-convex | ✗ | $\mathcal{O}\left(\frac{dS_\alpha F_0}{\varepsilon^2}\right)$ | ✗ | $\mathcal{O}\left(\frac{dLF_0}{\varepsilon^2}\right)$ | $\tilde{\mathcal{O}}\left(\frac{S_\alpha F_0}{\varepsilon^2}\right)$ | |
| Convex | $\mathcal{O}\left(\frac{S_\alpha R_{[1-\alpha]}^2}{\varepsilon}\right)$ | $\mathcal{O}\left(\frac{dS_\alpha R_{[1-\alpha]}^2}{\varepsilon}\log\frac{1}{\varepsilon}\right)$ | $\mathcal{O}\left(\frac{dLR^2}{\varepsilon}\right)$ | ✗ | $\tilde{\mathcal{O}}\left(\frac{S_\alpha R_{[1-\alpha]}^2}{\varepsilon}\right)$ | |
| Strongly convex | $\mathcal{O}\left(\frac{S_\alpha}{\mu_{1-\alpha}}\log\frac{1}{\varepsilon}\right)$ | $\mathcal{O}\left(\frac{S_\alpha}{\mu_{1-\alpha}}\log\frac{1}{\varepsilon}\right)$ | $\mathcal{O}\left(d\frac{L}{\mu}\log\frac{1}{\varepsilon}\right)$ | ✗ | $\tilde{\mathcal{O}}\left(\frac{S_\alpha}{\mu_{1-\alpha}}\log\frac{1}{\varepsilon}\right)$ | $\tilde{\mathcal{O}}\left(\frac{S_{\alpha/2}}{\sqrt{\mu_{1-\alpha}}}\log\frac{1}{\varepsilon}\right)$ |
| Order Oracle? | ✗ | ✓ | ✓ | ✓ | ✓ | ✓ |
| Acceleration? | ✗ | ✗ | ✗ | ✗ | ✗ | ✓ |

In this paper, we consider concept of zero-order oracle, termed *Order Oracle*, to address problem (1):

$$\phi(x, y) = \text{sign}\left[f(x) - f(y) + \delta(x, y)\right], \tag{2}$$

where $|\delta(x, y)| \leq \Delta$ is some bounded noise. The Order Oracle has the capability to compare two functions; however, in contrast to the zero-order oracle, it lacks the ability to calculate or utilize the actual value of the objective function. This concept closely mirrors the challenges encountered in real-world black-box optimization problems.

The motivation for the proposed oracle concept becomes more evident when considering the ongoing developments in generative models. Companies like Valio and SberAI have already embraced the active involvement of AI in dessert creation. Notably, Valio, as illustrated in Figure 1[*], employed AI to determine the optimal concentration of milk for chocolate bars based on available data. However, envision a scenario where AI creates customized chocolate for an individual by adjusting the concentration of ingredients. In such a case, a flavor comparison procedure, depicted in one iteration, would involve determining the preference *order*. Given that tastes can be closely aligned, introducing bounded noise in oracle (2) mitigates the potential for errors.

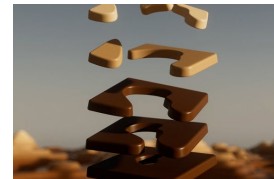

Figure 1: Valio's chocolate

To address the initial deterministic problem (1) with the Order Oracle (2), we propose a novel approach to algorithm design that uses class of coordinate descent methods (CD) Bubeck et al. (2015) as an optimization "tool" to integrate our oracle into multidimensional optimization. This approach is good in that we use linear search (where the Order Oracle is directly used) to determine not only the iteration step size, but also the gradient coordinate of the objective function. Thus demonstrating that the proposed algorithms are also adaptive. And for solving the initial stochastic problem (1) with the Order Oracle, we propose an approach based on normalized SGD, providing asymptotic convergence.

## 1.1 Our contributions

More specifically, our contributions are the following:

- We provide a novel approach to design algorithms for solving deterministic optimization problems (1) with Order Oracle (2) that achieves SOTA convergence results up to logarithm factor in the non-convex, convex, and strongly convex settings (see Table 1 and Algorithm 1).

- By using the approach proposed in this paper to create algorithms for solving deterministic optimization problem (1) with the Order Oracle (2), we have shown, on an example of strongly convex functions, that acceleration in such an oracle concept exists (see Algorithm 2 and Theorem 4.1). Moreover, we have shown how the convergence results of the accelerated algorithm can be improved when the problem is low-dimensional (the algorithm described in Appendix G shows convergence that even the Ellipsoid method cannot).

- We provide the first algorithm for solving a problem (1) with the stochastic Order Oracle concept, where the *order* between two functions on the same realization is determined.

- Through numerical experiments (see Section 7), we validate our theoretical results by comparing with first-order algorithms, as well as providing practical recommendations for implementing the first accelerated algorithm with the Order Oracle (2) (see Algorithm 2).

---

[*]The picture is taken from the company's Official Website. See more motivation in Appendix A

## 1.2 Main assumptions and notations

Before discussing related works, we present the notation and main assumptions we use in our work.

**Notation.** We use $\langle x, y \rangle := \sum_{i=1}^{d} x_i y_i$ to denote standard inner product of $x, y \in \mathbb{R}^d$. We denote Euclidean norm in $\mathbb{R}^d$ as $\|x\| := \sqrt{\sum_{i=1}^{d} x_i^2}$. In particular, this norm $\|x\| := \sqrt{\langle x, x \rangle}$ is related to the inner product. We use $\mathbf{e}_i \in \mathbb{R}^d$ to denote the $i$-th unit vector. We define the norms $\|x\|_{[\alpha]} := \sqrt{\sum_{i=1}^{d} L_i^\alpha x_i^2}$ and $\|x\|_{[\alpha]}^* := \sqrt{\sum_{i=1}^{d} \frac{1}{L_i^\alpha} x_i^2}$. We denote by $\nabla f(x)$ the full gradient of function $f$ at point $x \in \mathbb{R}^d$, and by $\nabla_i f(x)$ the $i$-th coordinate gradient. The sum of constant $L_i$ denotes as $S_\alpha := \sum_{i}^{d} L_i^\alpha$. We use $S^d(r) := \{x \in \mathbb{R}^d : \|x\| = r\}$ to denote Euclidean sphere. We use $\tilde{O}(\cdot)$ to hide the logarithmic coefficients. We denote $f^* := f(x^*)$ as the solution to initial problem.

For all our theoretical results, we assume that $f(x)$ is $L_i$-smooth with respect to its $i$-th coordinate:

**Assumption 1.1** (Smoothness). A function $f : \mathbb{R}^d \to \mathbb{R}$ is $L$-coordinate-Lipschitz for $L_1, L_2, ..., L_d > 0$ if for any $i \in [d], x \in \mathbb{R}^d$ and $h \in \mathbb{R}$ the following inequality holds:

$$|\nabla_i f(x + h\mathbf{e}_i) - \nabla_i f(x)| \le L_i |h|.$$

The smoothness assumption of the function is widely used in the optimization literature (e.g. Boyd and Vandenberghe, 2004; Nesterov et al., 2018). However, Assumption 1.1 is specific and frequently utilized in the context of optimization via the CD method Lin et al. (2014); Zhang and Xiao (2017); Mangold et al. (2023). This assumption means that for every input point $x$, if we alter its $i$-th coordinate by at most $h$, then the corresponding gradient $\nabla_i f(x + h\mathbf{e}_i)$ differs from $\nabla_i f(x)$ by at most $L_i$ times $|h|$.

Throughout this paper we assume that function $f(x)$ can be (strongly) convex w.r.t. the norm $\|\cdot\|_{[1-\alpha]}$:

**Assumption 1.2.** A function $f : \mathbb{R}^d \to \mathbb{R}$ is $\mu_{1-\alpha} \ge 0$ strongly convex w.r.t. the norm $\|\cdot\|_{[1-\alpha]}$ if for any $x, y \in \mathbb{R}^d$ the following inequality holds:

$$f(y) \ge f(x) + \langle \nabla f(x), y - x \rangle + \frac{\mu_{1-\alpha}}{2} \|y - x\|_{[1-\alpha]}^2.$$

Many works in convex optimization rely on the (strong) convexity assumption, as evidenced by references such as Duchi (2016); Shi et al. (2019); Asi et al. (2021). Since our work employs norms $\|\cdot\|_{[1-\alpha]}$ to derive theoretical estimates, Assumption 1.2 deviates slightly from the standard. However, this assumption of $\mu_{1-\alpha}$ (strong) convexity is extensively used in the following literature: Nesterov (2012); Lee and Sidford (2013); Allen-Zhu et al. (2016). Assumption 1.2 can conform to standard form of $\mu$ (strong) convexity via Euclidean norm if $\alpha = 1$. This assumption is convex provided $\mu_{1-\alpha} = 0$.

## 1.3 Paper organization

Further, this paper has the following structure. In Section 2, we provide a related work discussion. We present and analyze the non-accelerated algorithm in Section 3. Then in Section 4, we discuss the feasibility of accelerating the proposed algorithm in the concept of Order Oracle and provide theoretical guarantees. We provide the first algorithm that utilizes the stochastic concept of the Order Oracle in Section 5. In Section 6, we discuss the current results of this work. Through numerical experiments in Section 7, we validate the theoretical results. While Section 8 concludes the paper.

## 2 Related Works

The literature in the field of optimization is already extensive and continuously expanding, encompassing various problem formulations and assumptions. In this section, we provide an overview of the most relevant contributions to our work. Namely both CD methods and algorithms with Order Oracle.

**Algorithms with Order Oracle.** In the field of black-box optimization problem, special attention has been paid to algorithms that use only the Order Oracle. For example, Stochastic Three Points Method was proposed in Bergou et al. (2020), which uses an oracle that compares three function values at once and achieves the following oracle complexity in the strongly convex case $\mathcal{O}(dL/\mu \log 1/\varepsilon)$. A little later, the authors of Gorbunov et al. (2019) modified the Stochastic Three Points Method to the

case of importance sampling, improving the oracle complexity estimate in the strongly convex case $\mathcal{O}\left(S_\alpha/\mu \log 1/\varepsilon\right)$. Already in 2021, Saha et al. (2021) provided another algorithm in which the oracle compares two function values only once per iteration. The analysis of this algorithm is based on the Sign SGD and achieves oracle complexity in the strongly convex case $\mathcal{O}\left(dL/\mu \log 1/\varepsilon\right)$. The work of Tang et al. (2023) showed that the Order Oracle is also extensively used in *Reinforcement Learning with Human Feedback*, providing only in the non-convex case an estimate on oracle complexity $\mathcal{O}\left(d/\varepsilon^2\right)$. In this paper, we propose an alternative approach for creating optimization algorithms (via line search method), which achieves SOTA convergence results with logarithm accuracy in a class of non-accelerated algorithms, and provide the first accelerated algorithm with the Order Oracle.

**Coordinate descent (CD) methods.** In addition to full-gradient algorithms for smooth first-order optimization, coordinate descent algorithms are categorized into accelerated and non-accelerated algorithms. Non-accelerated algorithms typically converge at rates $1/\varepsilon$ and $1/\mu$ in convex ($\mu = 0$) and strongly convex ($\mu > 0$) cases, respectively, while accelerated algorithms achieve rates $1/\sqrt{\varepsilon}$ and $1/\sqrt{\mu}$ in convex and strongly convex cases, respectively. This classification dates back to 1983 when Nesterov introduced the optimal convergence rates for first-order algorithms Nesterov (1983). However, the fundamental difference between coordinate descent and full-gradient descent lies in the step taken along the $i$-th coordinate of the gradient (directional derivative). Monograph Bubeck et al. (2015) has demonstrated that if the direction is chosen uniformly, coordinate descent may require up to $d$ times more iterations than full-gradient descent. However, authors in Nesterov (2012) have shown that considering smoothness along the direction $L_i$ can improve the number of iterations $\mathcal{O}\left(S_\alpha R_{[1-\alpha]}^2/\varepsilon\right)$ and $\mathcal{O}\left(S_\alpha/\mu_{1-\alpha} \log \frac{1}{\varepsilon}\right)$ in convex and strongly convex cases, respectively, where $R_{[1-\alpha]} = \sup_{x \in \mathbb{R}^d: f(x) \leq f(x_0)} \|x - x^*\|_{[1-\alpha]}$ for $\alpha \in [0, 1]$. In the same paper Nesterov (2012), the authors demonstrated the potential for acceleration in coordinate descent through the scheme proposed in Nesterov (1983). Subsequently, in Lee and Sidford (2013), they analyzed and presented an accelerated version of coordinate descent (ACDM), along with the corresponding number of required $\mathcal{O}\left(\sqrt{dS_\alpha R_{[1-\alpha]}^2/\varepsilon}\right)$ and $\mathcal{O}\left(\sqrt{dS_\alpha/\mu_{1-\alpha}} \log \frac{1}{\varepsilon}\right)$ iterations in convex and strongly convex cases, respectively. It is noteworthy that, at that time, this iteration complexity was deemed unimprovable. However, a few years later, both Nesterov and Stich (2017) and Allen-Zhu et al. (2016) independently provided same results, demonstrating that it is indeed possible to enhance the iteration complexity for accelerated coordinate descent methods by modifying the probability of choosing the $i$-th coordinate: $L_i^{\alpha/2}/S_{\alpha/2}$. In this paper, we propose a coordinate descent algorithm with an Order Oracle (2), demonstrating that it achieves the same iteration complexity as first-order algorithms. Furthermore, based on accelerated coordinate descent Nesterov and Stich (2017), we establish on the strongly convex case that even with an Order Oracle, acceleration can be attained, resulting in the most favorable estimates on iteration complexity known to date.

## 3 Non-Accelerated Methods

In this section, we begin to present our main results, by introducing algorithms tailored to address the initial problem (1) utilizing the Order Oracle (2). Given the demand for this oracle concept, we furnish convergence guarantees and conduct comparisons with algorithms employing alternative oracles.

Our approach for developing a new algorithm involves incorporating the Order Oracle into an existing optimization method using linear search. We opt for linear search as the integration tool due to its compatibility with the Order Oracle concept. However, the choice of optimization method to which this linear search can be applied poses a crucial question. After careful consideration, we determined that the *coordinate descent method* (CDM) is the most suitable candidate. In each iteration, the CDM, given a step size $\zeta_k > 0$ and starting point $x_0 \in \mathbb{R}^d$, proceeds as follows:

$$x_{k+1} = x_k - \underbrace{\zeta_k \nabla_{i_k} f(x_k)}_{\eta_k} \mathbf{e}_{i_k}, \tag{3}$$

where $i_k$ is the coordinate index drawn from $[d]$. Note that the coordinate descent method step (3) requires both a step size $\zeta_k$ and a gradient coordinate value $\nabla_{i_k} f(x_k)$, which are scalars. Thus, by employing linear search $\eta_k = \arg\min_\eta \{f(x_k + \eta \mathbf{e}_{i_k})\}$ at each iteration, we can optimally determine both the direction of steepest descent and the step size to traverse along this direction, resulting in a fully adaptive algorithm. The next consideration is the strategy for coordinate selection at each iteration. Following the trend initiated by Nesterov (2012), we use a more general sampling distribution

than the uniform one, obtaining already *random coordinate descent* (RCD). Specifically, for $\alpha > 0$, we assume that a random generator $\mathcal{R}_\alpha(L)$ independently selects $i_k$ from the following distribution:

$$p_\alpha(i) = L_i^\alpha / S_\alpha, \quad i \in [d]. \tag{4}$$

We are now ready to introduce a method designed to solve problem (1) utilizing the oracle concept (2).

This algorithm falls under the category of coordinate methods and is named *random coordinate descent with order oracle* (OrderRCD), see Algorithm 1. It should be noted that the inherent "stochasticity" of problem (1) is artificially induced by randomized procedure used to select the $i$-th coordinate (4). Here, $\xi$ denotes the $i$-th coordinate, and $\mathcal{D}$ represents distribution $p_\alpha(i)$ from (4). The *golden ratio method* (GRM) serves

---

**Algorithm 1** Random Coordinate Descent with Order Oracle (OrderRCD)

---

**Input:** $x_0 \in \mathbb{R}^d$, random generator $\mathcal{R}_\alpha(L)$
**for** $k = 0$ **to** $N - 1$ **do**
  1.  choose active coordinate $i_k = \mathcal{R}_\alpha(L)$
  2.  compute $\eta_k = \mathrm{argmin}_\eta \{f(x_k + \eta \mathbf{e}_{i_k})\}$ via (GRM)
  3.  $x_{k+1} \leftarrow x_k + \eta_k \mathbf{e}_{i_k}$
**end for**
**Return:** $x_N$

---

as the linear search algorithm, which is where the Order Oracle is used. It is known that GRM (which is described in Appendix C.2) requires $N = \mathcal{O}(\log 1/\epsilon)$ iterations to achieve the desired accuracy $\epsilon$ (in terms of function) of the solution to the linear search problem (namely, $\eta_k = \mathrm{argmin}_\eta \{f(x_k + \eta \mathbf{e}_{i_k})\}$).

Next, we present our theoretical results, demonstrating through convergence analyses that random coordinate descent with order oracle (OrderRCD) exhibits competitive iteration complexity compared to *first-order algorithms* when applied to *non-convex*, *convex*, or *strongly convex* functions.

## 3.1 Non-convex setting

**Theorem 3.1** (non-convex). *Let function f(x) satisfies Assumption 1.1, $N$ is the number of iterations, $F_0 = f(x_0) - f(x^*)$, then Algorithm 1 (OrderRCD) with oracle (2) guarantees an error:*

$$\frac{1}{N} \sum_{k=0}^{N-1} \left( \|\nabla f(x_k)\|_{[1-\alpha]}^* \right)^2 \leq \mathcal{O}\left( \frac{S_\alpha F_0}{N} + S_\alpha \epsilon + S_\alpha \Phi \Delta \right),$$

*where $\epsilon$ is the accuracy of solving linear search problem by function and $\Phi = \frac{1+\sqrt{5}}{2}$ is golden ratio.*

The convergence results of Theorem 3.1 imply the existence of a point $k \in [N]$ where $(\|\nabla f(x_k)\|_{[1-\alpha]}^*)^2 \leq \mathcal{O}\left(\frac{S_\alpha F_0}{N}\right)$ holds true. Additionally, Theorem 3.1 indicate that, to achieve the desired accuracy $\varepsilon$ (according to the gradient norm), Algorithm 1 requires $N = \mathcal{O}\left(S_\alpha F_0/\varepsilon^2\right)$ iterations and $T = \tilde{\mathcal{O}}\left(S_\alpha F_0/\varepsilon^2\right)$ calls to the oracle (2). The maximum admissible noise level ensuring the desired accuracy should not exceed $\sim \varepsilon^2$. Notably, the convergence rate of random coordinate descent with the order oracle (OrderRCD), assuming minimal noise ($\Delta \leq \varepsilon^2$), is equal to *first-order method*: random coordinate descent (RCD). It is also characteristic (within the coordinate methods class) to be inferior to *gradient descent* (Ghadimi and Lan, 2013, under $L$ smoothness assumption). For instance, in the case when $\alpha = 1$, the convergence rates $\mathcal{O}(LF_0/N)$ is better because $L \leq \sum_{i=1}^d L_i$. Not surprisingly, in terms of oracle complexity, Algorithm (1) is logarithmically inferior to first-order algorithms. This discrepancy reflects the "cost" associated with employing GRM, where the $\tilde{O}(\cdot)$ to hide the *logarithmic coefficient* representing the number of oracle calls in the GRM, *contingent upon its accuracy $\epsilon$*. A detailed proof of Theorem 3.1 is given in Appendix D.1.

## 3.2 Convex setting

We now prove an analogous theorem on the convergence of the algorithm when the function is convex, i.e., additionally assuming that Assumption 1.2 is satisfied with $\mu_{1-\alpha} = 0$.

**Theorem 3.2** (convex). *Let function $f(x)$ satisfies Assumption 1.1 (L-Smoothness) and Assumption 1.2 (convexity, $\mu_{1-\alpha} = 0$), then Algorithm 1 (OrderRCD) with oracle (2) guarantees an error:*

$$\mathbb{E}[f(x_N)] - f(x^*) \leq \mathcal{O}\left( \frac{S_\alpha R_{[1-\alpha]}^2}{N} + \frac{2 S_\alpha R_{[1-\alpha]}^2 (\epsilon + \Phi \Delta)}{F_{N-1}} \right),$$

*where $\epsilon$ is an inner problem accuracy, $R_{[1-\alpha]} = \sup_{x \in \mathbb{R}^d : f(x) \leq f(x_0)} \|x - x^*\|_{[1-\alpha]}$, and $\Phi = \frac{1+\sqrt{5}}{2}$.*

In comparison to Theorem 3.1 (non-convex setting), the convergence results of Algorithm 1 demonstrate improvement under the assumption of function convexity (Assumption 1.2 with $\mu_{1-\alpha} = 0$). Specifically, according to Theorem 3.2, random coordinate descent with order oracle (OrderRCD) requires $N = \mathcal{O}(S_\alpha R_{[1-\alpha]}^2/\varepsilon)$ iterations and $T = \tilde{\mathcal{O}}(S_\alpha R_{[1-\alpha]}^2/\varepsilon)$ oracle calls to achieve the desired accuracy $\varepsilon$ (where $\mathbb{E}\left[f(x_N)\right] - f(x^*) \leq \varepsilon$). However, in cases where $\Delta > 0$, the condition for maximum noise remains unchanged compared to the non-convex setting. Furthermore, the iteration complexity $N$ aligns with that of the first-order algorithm, (*random coordinate descent* (RCD) Nesterov, 2012). Similarly, akin to Theorem 3.2, the convergence rate of *gradient descent* (Nesterov et al., 2018, under assuming $L$ smoothness and convexity) outperforms that of Algorithm 1. Moreover, when $\alpha = 0$ (corresponding to a *uniform distribution* with probability $p_\alpha(i) = 1/d$), OrderRCD (like all coordinate methods) necessitates $d$ times more iterations than gradient descent. Concerning oracle complexity $T$, a logarithmic coefficient is evident, correlating with the number of oracle calls per iteration of Algorithm 1, line 2. For a comprehensive proof of Theorem 3.2, refer to Appendix D.2.

### 3.3 Strongly convex setting

In this section we consider case when function is strongly convex (see Assumption 1.2, $\mu_{1-\alpha} > 0$).

**Theorem 3.3** (strongly convex). *Let function $f(x)$ satisfies Assumption 1.1 (L-Smoothness) and Assumption 1.2 (convexity, $\mu_{1-\alpha} > 0$), then Algorithm 1 with oracle (2) has a linear convergence rate:*

$$\mathbb{E}\left[f(x_N)\right] - f(x^*) \leq \left(1 - \frac{\mu_{1-\alpha}}{S_\alpha}\right)^N F_0 + \frac{2S_\alpha \epsilon}{\mu_{1-\alpha}} + \frac{2cS_\alpha \Phi \Delta}{\mu_{1-\alpha}},$$

*where $c$ is some constant, $\epsilon$ is the GRM accuracy (by function) and $\Phi = \frac{1+\sqrt{5}}{2}$ is golden ratio.*

As depicted in Theorem 3.3 Algorithm 1 exhibits a linear convergence rate, achieving the desired accuracy $\varepsilon$ in $N = \mathcal{O}\left(S_\alpha/\mu_{1-\alpha} \log \frac{1}{\varepsilon}\right)$ iterations and $T = \tilde{\mathcal{O}}\left(S_\alpha/\mu_{1-\alpha} \log \frac{1}{\varepsilon}\right)$ oracle calls. Furthermore, there's an enhancement in the maximum noise level $\Delta$, reaching $\sim \mu_{1-\alpha}\varepsilon$. It's notable that a weaker strong convexity condition was employed in the proof of Theorem 3.3 (see Appendix D.3):

$$\|\nabla f(x)\|_{[1-\alpha]}^2 \geq 2\mu_{1-\alpha}(f(x) - f(x^*)), \forall x \in \mathbb{R}^d. \tag{5}$$

This condition (in the case $\alpha = 1$), also known as the Polyak–Lojasiewicz or Gradient-dominated functions condition Polyak (1963); Lojasiewicz (1963); Karimi et al. (2016); Belkin (2021), encompasses a broad class of functions: convex functions, strongly convex functions, sum of squares (e.g. where considering a system of non-linear equations), invex and *non-convex functions*, as well as over-parameterized systems. Also it is shown in Yue et al. (2023) that *non-accelerated algorithms (like Algorithm 1) are optimal for L-smooth* problems under the Polyak–Lojasiewicz condition (5).

**High probability deviations bounds.** Given that OrderRCD method in strongly convex setting demonstrates a linear convergence rate and employs a randomization in coordinate selection, we can derive exact estimates of *high deviation probabilities* using Markov's inequality Anikin et al. (2015):

$$\mathcal{P}\left(f(x_{N(\varepsilon\sigma)}) - f^* \geq \varepsilon\right) \leq \sigma \frac{\mathbb{E}\left[f(x_{N(\varepsilon\sigma)}] - f^*\right.}{\varepsilon\sigma} \leq \sigma.$$

## 4 Accelerated Method

Random coordinate descent with order oracle (OrderRCD) demonstrates efficiency in the class of coordinate methods. The number of iterations $N$ required to achieve the desired accuracy $\varepsilon$ is fully identical to random coordinate descent (RCD), which is the best among the non-accelerated methods in this class, and are also not inferior to existing competitors with Order Oracle. This fact confirms that our proposed approach to developing a novel optimization algorithm is successful. However, Algorithm 1 is still not optimal because it belongs to non-accelerated algorithms. Nevertheless, Section 3 gives hope for the possibility of acceleration among algorithms using the oracle concept (2).

In this section, we demonstrate on the example of the strongly convex (Assumption 1.2 is satisfied with constant $\mu_{1-\alpha} > 0$) problem (1) that *acceleration* in the class of optimization algorithms using the Order Oracle (2) *exists*! For simplicity, we consider the case when $\Delta = 0$.

Our approach to creating an accelerated algorithm closely mirrors the one proposed in Section 3, which involves adapting an existing optimization method (from the class of coordinate algorithms) to the Order Oracle using linear search method (namely, golden ratio method, GRM). Among the accelerated algorithms in the class of coordinate descent, two stand out: (*accelerated coordinate descent method* (ACDM), Nesterov and Stich, 2017) and (*accelerated coordinate descent method with non-uniform sampling* (NU-ACDM), Allen-Zhu et al., 2016). These algorithms boast the same convergence rate and are considered among the best available today. Therefore, we select one of them (specifically, ACDM) as the base algorithm to adapt to our oracle concept (2). At each iteration, the ACDM, given parameters such as $\alpha_k$, $\beta_k$, $a_{k+1}$, $A_{k+1}$, $B_{k+1}$, Lipschitz coordinate constant $L_i$, strong convexity $\mu_{1-\alpha}$, distribution $p_\beta(i)$, and a starting point $x_0 = z_0$, proceeds as follows:

$$y_k = (1 - \alpha_k)x_k + \alpha_k w_k,$$
$$x_{k+1} = y_k - (1/L_{i_k})\nabla_{i_k}f(y_k)\mathbf{e}_{i_k}, \tag{6}$$
$$z_{k+1} = w_k - \frac{a_{k+1}}{L_{i_k}^{1-\alpha}B_{k+1}p_\beta(i)}\nabla_{i_k}f(y_k)\mathbf{e}_{i_k}, \tag{7}$$

where $w_k = (1 - \beta_k)z_k + \beta_k y_k$. Looking at the $x_{k+1}$ update (6), it seems that it should not be difficult to determine the step size $1/L_{i_k}$ and the value of the gradient coordinate $\nabla_{i_k}f(y_k)$ using a linear search, as demonstrated in Algorithm 1. However, substituting the same $\eta_k$ instead of $(1/L_{i_k})\nabla_{i_k}f(y_k)$ into the $z_{k+1}$ update (7) isn't straightforward. The step with linear search is larger than the original step of $(1/L_{i_k})\nabla_{i_k}f(y_k)$, potentially leading to a paradoxical situation where the step worsens with linear search. This is because there is no guarantee that the function $f$ is monotonically decreasing along the sequences $\{z_k\}_{k=0}^\infty$ and $\{x_k\}_{k=0}^\infty$ Nesterov (1983). Nevertheless, we have successfully addressed this challenge and we are ready to present the first accelerated algorithm utilizing only the Order Oracle: the *accelerated coordinate descent method with order oracle.*

It is evident from Algorithm 2 that the challenge was addressed by incorporating a secondary linear search, ensuring that the update step in $z_{k+1}$ with linear search is at least as effective as (7). However, unlike Algorithm 1, OrderACDM cannot be deemed fully adaptive as it necessitates knowledge of the strong convexity constant $\mu_{1-\alpha}$ and the Smoothness constant $L_i$ (which disappears in the case where $\alpha = 0$). Despite this, we are ready to present the main advantage of Algorithm: Faster convergence rate of Accelerated Coordinate Descent Method with Order Oracle.

---

**Algorithm 2** Accelerated Coordinate Descent Method with Order Oracle (OrderACDM)

---

**Input:** $x_0 = z_0 \in \mathbb{R}^d$, $\mathcal{R}_\alpha(L)$, $A_0 = 0$, $B_0 = 1$, $\beta = \frac{\alpha}{2}$
**for** $k = 0$ **to** $N - 1$ **do**
1. choose active coordinate $i_k = \mathcal{R}_\beta(L)$
2. find parameter $a_{k+1}$ from $a_{k+1}^2 S_\beta^2 = A_{k+1}B_{k+1}$, where $A_{k+1} = A_k + a_{k+1}$ and
$$B_{k+1} = B_k + \mu_{1-\alpha}a_{k+1}$$
3. $\alpha_k \leftarrow \frac{a_{k+1}}{A_{k+1}}$
4. $\beta_k \leftarrow \frac{\mu_{1-\alpha}a_{k+1}}{B_{k+1}}$
5. $y_k \leftarrow \frac{(1-\alpha_k)x_k + \alpha_k(1-\beta_k)z_k}{1-\alpha_k\beta_k}$
6. compute $\eta_k = \arg\min_\eta\{f(y_k + \eta\mathbf{e}_{i_k})\}$ via (GRM)
7. $x_{k+1} \leftarrow y_k + \eta_k\mathbf{e}_{i_k}$
8. $w_k \leftarrow (1 - \beta_k)z_k + \beta_k y_k + \frac{a_{k+1}L_{i_k}^\alpha}{B_{k+1}p_\beta(i)}\eta_k\mathbf{e}_{i_k}$
9. compute $\zeta_k = \arg\min_\zeta\{f(w_k + \zeta\mathbf{e}_{i_k})\}$ via (GRM)
10. $z_{k+1} \leftarrow w_k + \zeta_k\mathbf{e}_{i_k}$
**end for**
**Return:** $x_N$

---

**Theorem 4.1.** *Let function $f(x)$ is strongly convex (Assumption 1.2), $L$-Smoothness (Assumption 1.1) and $L_{[1-\alpha]}$-Smoothness[†], then Algorithm 2 (OrderACDM) with oracle (2) guarantees an error*

$$\mathbb{E}[f(x_N)] - f(x^*) \leq \left(1 - \frac{\sqrt{\mu_{1-\alpha}}}{S_{\alpha/2}}\right)^N F_0,$$

*where $F_0 = f(x_0) - f(x^*)$, $S_{\alpha/2} = \sum_{i=1}^d L_i^{\alpha/2}$.*

Compared to the results of Section 3, the convergence rate demonstrated in Theorem 4.1 is superior, *confirming the potential for acceleration* in algorithms utilizing the Order Oracle. To achieve the desired accuracy $\varepsilon$, Algorithm 2 (OrderACDM) requires $N = \mathcal{O}\left(S_{\alpha/2}/\sqrt{\mu_{1-\alpha}}\log\frac{1}{\varepsilon}\right)$ iterations

---

[†]We assume that for any $x, y \in \mathbb{R}^d$ it holds: $f(y) \leq f(x) + \langle\nabla f(x), y - x\rangle + \frac{L_{[1-\alpha]}}{2}\|y - x\|_{[1-\alpha]}^2$.

and $T = \tilde{\mathcal{O}}\left(S_{\alpha/2}/\sqrt{\mu_{1-\alpha}}\log\frac{1}{\varepsilon}\right)$ oracle calls, significantly outperforming existing competitors that utilize the Order Oracle (see Table 1). One notable difference from Algorithm 1 is selection of the active coordinate $i = \mathcal{R}_\beta(L)$, where $\beta = \alpha/2$. This choice aims to eliminate dimensionality $d$ from the iteration and oracle complexities derived in Lee and Sidford (2013), since $S_{\alpha/2} \leq \sqrt{dS_\alpha}$. A detailed proof of Theorem 4.1 is given in Appendix E.

## 5 Stochastic Order Oracle Concept

This section is devoted to an equally important concept of the Order Oracle (2), namely its modification to the stochastic case, where the values of two functions on one realization $\xi$ are compared:

$$\phi(x, y, \xi) = \text{sign}\left[f(x, \xi) - f(y, \xi)\right]. \tag{8}$$

For simplicity, we assume that the oracle (8) is not subject to adversarial noise. This oracle concept can also be motivated by the creation of an ideal chocolate only for a group of people on average. That is, in this concept the $\xi$-th realization of the function can be understood as the $\xi$-th individual of group.

To address the stochastic black-box optimization problem (1) when only the Order Oracle (8) is available, we provide the first optimization algorithm that uses exactly the stochastic oracle concept (8):

$$x_{k+1} = x_k - \eta_k \phi(x_k + \gamma_k \mathbf{e}_k, x_k - \gamma_k \mathbf{e}_k, \xi_k)\mathbf{e}_k, \tag{9}$$

where $\gamma_k > 0$ is a smoothing parameter, $\mathbf{e}_k \in S^d(1)$ is a vector uniformly distributed on the Euclidean sphere. In order to proceed to convergence guarantees of this method (9), we use the auxiliary results.

**Lemma 5.1.** *Let function $f$ be $L$-smooth*[‡], $\gamma_k = \frac{\|\nabla f(x_k, \xi_k)\|}{\sqrt{d}L}$, $\mathbf{e}_k \in S^d(1)$, *then the following holds:*

$$\phi(x_k + \gamma_k \mathbf{e}_k, x_k - \gamma_k \mathbf{e}_k, \xi_k)\mathbf{e}_k = \text{sign}\left[\langle \nabla f(x_k, \xi_k), \mathbf{e}_k \rangle\right]\mathbf{e}_k.$$

Now using Lemma 5.1 we show how our algorithm (9) with oracle (8) is related to normalized SGD.

**Lemma 5.2.** *Let vector $\nabla f(x, \xi) \in \mathbb{R}^d$ and vector $\mathbf{e} \in S^d(1)$, then with some constant $c$ we have*

$$\mathbb{E}_{\mathbf{e}}\left[\text{sign}\left[\langle \nabla f(x, \xi), \mathbf{e} \rangle\right]\mathbf{e}\right] = \frac{c}{\sqrt{d}} \cdot \frac{\nabla f(x, \xi)}{\|\nabla f(x, \xi)\|}.$$

As Lemma 5.2 shows, the direction in which a step is taken in Algorithm (9) is the direction of normalized stochastic gradient descent. Given this fact and based on the work of Polyak and Tsypkin (1980) we can provide asymptotic convergence for the first algorithm using the oracle concept (8).

**Theorem 5.3** (Asymptotic convergence). *Let the function $f$ be a $L$-smooth and $\mathbf{e}_k \in S^d(1)$, then for the algorithm (9) with step size $\eta_k = \eta/k$ the value $\sqrt{N}(x_k - x^*)$ is asymptotically normal: $\sqrt{N}(x_k - x^*) \sim \mathcal{N}(0, V)$, where the matrix $V$ is as follows:*

$$V = \frac{\eta^2}{d}\left(2\eta(1 - 1/d)\frac{c}{\sqrt{d}}\alpha\nabla^2 f(x^*) - I\right)^{-1},$$

$\alpha = \int \|z\|^{-1} dP(z) < \infty$, $2\eta(1 - 1/d)\frac{c}{\sqrt{d}}\alpha\nabla^2 f(x^*) > I$ *(where $I$ is unit matrix).*

For a detailed proof of Theorem 5.3, including a consideration of auxiliary Lemmas, see Appendix F.

## 6 Discussion

In Section 3, we showed that such a multidimensional optimization problem (1) can be solved using Algorithm 1, which utilizes only comparative information. The fact that such an algorithm exists is not surprising, but the fact that it is as good as other methods with this oracle or first-order algorithms (in the class of coordinate algorithms) is positive news that allows us to think about the optimality of the algorithm. In the Subsection 3.3, we showed that when the Polyak-Lojasiewicz condition

---

[‡]We assume that for any $x, y \in \mathbb{R}^d$ it holds: $f(y, \xi) \leq f(x, \xi) + \langle \nabla f(x, \xi), y - x \rangle + \frac{L}{2}\|y - x\|^2$.

is satisfied, we can perhaps consider that OrderRCD has optimal iteration complexity (see, Yue et al., 2023). However, when the other conditions are satisfied, this is not the case, since Algorithm 1 belongs to non-accelerated algorithms. Therefore, we believe that Section 4 provides a vector for the development of the question of optimality by showing an OrderACDM, which can perhaps be considered optimal in terms of iteration complexity. However, in the case of a *low-dimensional problem*, we can take a "Private communication" approach proposed Yurii Nesterov (for a more detailed description, see Appendix G) to create a more efficient algorithm using only oracle (2).

Before move to the numerical experiments, it's important to note that accelerated Algorithm 2 employs the golden ratio method (GRM) twice per iteration. This might raise concerns about its computational efficiency. However, in practice, we found that OrderACDM converges efficiently, comparable to the first-order algorithm, even when utilizing the golden ratio method only once. For a detailed analysis of the numerical experiments, refer to next Section 7 and Appendix B.

## 7 Numerical Experiments

In this section, we investigate the performance of the proposed algorithms in the corresponding Sections 3 and 4 on a numerical experiment. We compare OrderRCD (see Algorithm 1) and OrderACDM (see Algorithm 2), which utilize deterministic oracle concept (2) with existing first-order state-of-the-art algorithms. The goal is to highlight our theoretical results.
The optimization problem (1) has a standard quadratic form: $\min_{x \in \mathbb{R}^d} f(x) := \frac{1}{2} \langle x, Ax \rangle - \langle b, x \rangle + c$, where $A \in \mathbb{R}^{d \times d}$, $b \in \mathbb{R}^d$, and $c \in \mathbb{R}$. We use a uniform distribution ($\alpha = 0$) to choose the active coordinate, then the distribution (4) has the following form: $p_0(i) = 1/d$. For such a problem, the assumptions of $L$-Coordinate-Lipschitz smoothness (Assumption 1.1) and $\mu_{1-\alpha}$-strong convexity (Assumption 1.2) are satisfied, where $L_i = A_{ii}$. In all experiments, we employ the golden ratio method (GRM) to solve the linear search problem with a precision of $\epsilon = 10^{-8}$.

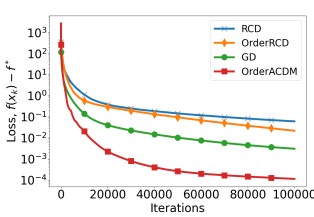

Figure 2: Comparison of algorithms proposed with non-accelerated first-order algorithms.

In Figure 2, we compare the convergence *random coordinate descent with order oracle* (OrderRCD) and *accelerated coordinate descent method with order oracle* (OrderACDM) with the SOTA non-accelerated algorithms: *random coordinate descent* (RCD) from Nesterov (2012), as well as *gradient descent* (GD). Non-accelerated coordinate algorithms, both for first-order oracle (RCD) and for our oracle concept (OrderRCD), are observed to lag behind gradient descent, confirming our theoretical derivations in Section 3. Interestingly, the random coordinate descent with order oracle even *outperforms its first-order counterpart*, despite the limitations associated with oracle usage (only Order Oracle (2) available). This observation can be attributed to the adaptiveness of Algorithm 1, as OrderRCD employs an exact step in the steepest descent direction obtained using the golden ratio method (GRM) at each iteration. Additionally, we can observe perhaps the most significant result demonstrated in Figure 2: *acceleration in our oracle concept* (2) *exists*! We see that accelerated coordinate descent method with order oracle outpaces the convergence speed of all non-accelerated algorithms, including first-order coordinate (RCD) and full-gradient (GD) methods. In this experiment, OrderACDM was implemented using the method described in Algorithm 2 with $\zeta_k = 0$ (i.e., with one golden ratio method); RCD and GD used a constant step size, specifically $1/L_i$ and $1/L$.

## 8 Conclusion

We proposed a new approach to design optimization algorithms using only the deterministic concept of Order Oracle (2) by providing theoretical guarantees (showing SOTA results up to logarithm factor) for non-accelerated algorithms in non-convex, convex and strongly convex settings. We also discussed under which condition the Algorithm 1 is optimal (under the Polyak-Lojasiewicz condition). Using the proposed approach, we have shown that acceleration in the deterministic concept of the Order Oracle exists, thereby opening up a whole range of potential research. Furthermore, we have shown how the evaluation of the accelerated algorithm (so still convex tuning) can be improved by considering low-dimensional problems. Moreover, we provided first-of-its-kind theoretical guarantees for an algorithm utilizing the stochastic concept of Order Oracle (8). Finally, we demonstrated the effectiveness of the proposed algorithms (OrderRCD and OrderACDM) on numerical experiments, thereby validating the theoretical results. We provided practical recommendations for implementation of these algorithms.

## 9   Authors' Affiliation Clarification

MIPT = *Moscow Institute of Physics and Technology, Dolgoprudny, Russia*;
Skoltech = *Skolkovo Institute of Science and Technology, Moscow, Russia*;
ISP RAS = *Ivannikov Institute for System Programming of the Russian Academy of Sciences, Moscow, Russia*;
Innopolis = *Research Center for Artificial Intelligence, Innopolis University, Innopolis, Russia*;
MI RAS = *Steklov Mathematical Institute of Russian Academy of Sciences, Moscow, Russia*.

## 10   Acknowledgments and Disclosure of Funding

This research has been financially supported by The Analytical Center for the Government of the Russian Federation (Agreement No. 70-2021-00143 01.11.2021, IGK 000000D730324P540002). The authors are grateful to Eduard Gorbunov, Andrey Neznamov, Alexander Vedyakhin.

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

# APPENDIX
# Acceleration Exists! Optimization Problems When Oracle Can Only Compare Objective Function Values

## A    A Few More Words on the Motivation Behind the Order Oracle Concept

In this Section, we would like to emphasize the motivation behind the problem statement discussed in this paper. In particular, we would like to demonstrate the applicability of this work.

### A.1    Perfect coffee for everyone

As already demonstrated in Section 1 (Introduction) with the example of chocolate, the deterministic concept of the Order Oracle has many potential applications. However, this research was initiated due to a challenge one of the co-authors faced during the realization of *a startup: the creation of an ideal coffee machine that can make the perfect drink for each customer*. This startup has just started its life cycle. At the moment we have designed a coffee machine that is functioning at the testing stage (see the photo of the machine in Figure 3 and the 3D model in Figure 4).

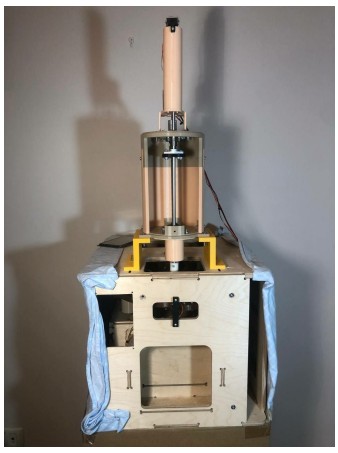

Figure 3: Smart coffee machine.

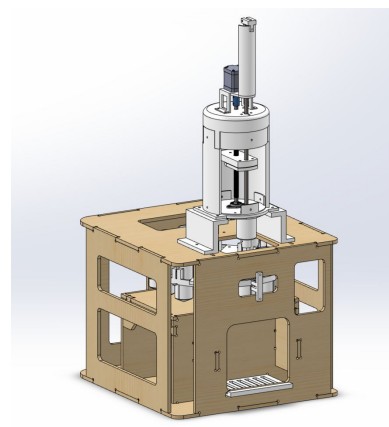

Figure 4: 3D model of a smart coffee machine.

**Brief description of the coffee machine.**    A coffee machine that can make the perfect coffee. By varying the proportions of strong Robusta beans, which give a bitter "Starbucks flavor", and mild Arabica beans, we can find the perfect level of bitterness and coffee strength. By varying the amount of milk and cream we can find the right level of milkiness and fat content. We can also adjust the amount of other ingredients such as sugar, ice, lemon juice, chocolate, different syrups to make sure that the customer will definitely like this coffee.

### A.2    Order Oracle: a zero-order oracle close to reality

In this Subsection, we would like to show that the oracle concept considered in this paper is perhaps the closest to reality.

One of the key criteria for evaluating optimization algorithms is oracle complexity. Oracles are commonly used in theoretical estimates, where they offer insights into the function's behavior. For

instance, the first-order oracle is prevalent in machine learning literature: the authors of Gorbunov et al. (2020); Gurbuzbalaban et al. (2021); Huang et al. (2022) use the oracle to obtain the gradient value $\nabla f(x_k)$ of a function at a given point $x_k$. And in Stich and Karimireddy (2020); Ajalloeian and Stich (2020); Glasgow et al. (2022), the authors assume some adversarial environment where the oracle is inaccurate, i.e., the oracle produces the gradient of the function at a given point with some adversarial noise (adversarial refers to the noise that accumulates over iterations). The authors of Nesterov (2012); Lee and Sidford (2013); Mangold et al. (2023) also refer their oracles that produce only the gradient coordinate of the function $\nabla_i f(x_k)$ at a given point $x_k$ to a first-order oracle. However, this oracle formally can also be referred to the so-called gradient-free oracle Gasnikov et al. (2022a), since the algorithm does not use the whole gradient, but only the directional derivative. The following works Jiang et al. (2019); Nesterov (2021); Agafonov et al. (2023) use a higher-order oracle to obtain information, for example, about the Hesse matrix of a function at a given point. But there are oracle concepts closer to reality, for example in Bach and Perchet (2016); Shamir (2017); Akhavan et al. (2021); Gasnikov et al. (2022b); Kornilov et al. (2023) the authors develop algorithms that use only information about the function value $f(x)$, possibly with some adversarial noise. Such a concept is very common in the field of black-box optimization. In this paper we consider a concept of zero-order oracle: the Order Oracle (2), which is even closer to reality, where even the function value is not available to us, but only the order between two functions (the possibility to compare). Moreover, we take into account the presence of noise (which seems to be natural in applied realities) in such an oracle.

### A.3 Why is a convex/concave function being considered?

In this Subsection, we would like to emphasize the organicity of considering the convexity/concave assumption of the objective function of the original problem (1).

"*In terms of the law of diminishing utility, the utility function is a concave function in ordinary coordinates:*" In economic science there is a concept of "utility". Usually this term is used when describing a consumer who makes a set of several goods (like a basket in a supermarket). Each such set has some utility for the consumer, and he tries to maximize it. We do not consider a consumer who makes a set of goods, but a consumer who makes the goods themselves from a set of their characteristics. We believe that here the consumer is already maximizing a "preference function" to emphasize the difference with the "utility function". For convenience, we consider all characteristics to be useful (the larger the diagonal of the TV set, the more the consumer likes it). "Harmful" characteristics, such as price, we simply replace with the inverse, because the inverse price $1/p$ will already be useful (among goods with the same characteristics, it is logical to assume that the consumer will choose the cheaper one). Gossen's first law sounds as follows: "The magnitude [intensity] of pleasure decreases continuously if we continue to satisfy one and the same enjoyment without interruption until satiety is ultimately reached" Kurz (2016). It is more commonly referred to as the law of diminishing marginal utility of goods. The decreasing marginal utility of a good actually means that the derivative or gradient of the utility function decreases as the quantity of the "useful attribute" increases. This leads us to the concavity of the utility function. It is therefore natural to assume that *the "preference function" will also be concave*.

## B Additional Numerical Experiments

In this Section, we provide additional experiments that solve the problem discussed in Section 7. We also give practical recommendations for implementing the accelerated Algorithm 2 in practice.

In Figure 5, we investigate effect of adversarial noise $\delta(x, y)$ from deterministic oracle concept (2) on a random coordinate descent with order oracle (OrderRCD). We used $(\delta(x, y) = \Delta \cdot \cos x \cdot \sin y)$ as the adversarial deterministic noise, where $\Delta$ (s.t. $|\delta(x, y)| \leq \Delta$) is the maximum noise level. In Figure 5, we see that the adversarial noise justifies its name as it accumulates over iterations. In addition, we observe that the convergence of the non-accelerated coordinate method depends directly on the maximum noise level $\Delta$, namely, the lower the noise level, the more accurately the algorithm converges. And finally we see that our theoretical results from Section 3 are confirmed, which indicate that the asymptote to which the algorithm converges can be controlled by the maximum noise level $\Delta$, for example, in this case, when $\alpha = 0$, the condition for achieving the desired accuracy $\varepsilon$ looks as

follows: $\Delta \leq \mu_{1-\alpha}\varepsilon/d$. Or we can rephrase this condition: the OrderRCD has a linear convergence rate to the asymptote depending on level noise $\Delta$.

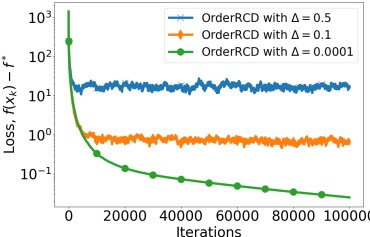

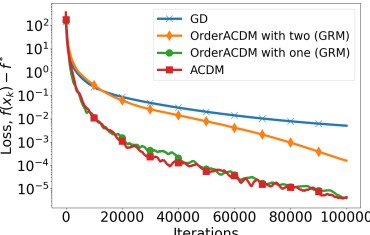

Figure 5: Effects of adversarial noise from the Order Oracle (2) on the convergence of random coordinate descent (OrderRCD). Here we optimize $f(x)$ with the parameters: $d = 100$ (dimensional of problem), $\Delta = \{0.5, 0.1, 0.0001\}$ (maximum noise level).

Figure 6: Effect of the second of golden ratio method on the convergence of accelerated coordinate descent with order oracle. Here we optimize $f(x)$ with the parameters: $d = 100$ (dimensional of problem), $\Delta = 0$ (maximum noise level in the Order Oracle).

In Figure 6, we illustrate the advantage of employing Algorithm 2 with one method of line search ($\zeta_k = 0$). We can observe that the method proposed in Section 4, the accelerated coordinate descent with order oracle (OrderACDM with two GRM), indeed qualifies as accelerated (thus affirming the theoretical findings of Section 4) as it outperforms gradient descent (GD), which can be considered a boundary between non-accelerated and accelerated coordinate methods. However, this algorithm significantly lags behind the accelerated coordinate descent method (ACDM, Nesterov and Stich, 2017). The reason may be the delayed momentum effect caused by using the golden ratio method (GRM) a second time (see line 10 of Algorithm 2). Addressing this may involve utilizing the golden section method only once per iteration in Algorithm 2 (that is, substitute $\zeta_k = 0$ into line 10). Indeed, we observe that when using the golden section method once per iteration in Algorithm 2, the OrderACDM enhances convergence rate and does not fall behind its first-order counterpart (ACDM) which confirms our theoretical results. That is why we recommend to utilize in practice accelerated coordinate descent method with order oracle (OrderACDM, see Algorithm 2) with only one the golden ratio method (GRM) per iteration.

**Technical Information.**    All experiments were performed on an INTEL CORE i5 2.10 GHz processor. The performance of each Figure depended on the particular algorithm, for example, *gradient descent* (GD) performed 40000 iterations in 0.1 second, while *random coordinate descent* (RCD) and *accelerated coordinate descent method* (ACDM) performed 40000 iterations in 0.2 seconds and 0.4 seconds respectively. But Algorithms 1 and 2 proposed in this paper, by virtue of using linear search at each iteration performed 40000 iterations in 02:02 minutes and 02:13 minutes respectively.

## C    Auxiliary Results

In this section we provide auxiliary materials that are used in the proof of Theorems.

### C.1    Basic inequalities and assumptions

**Basic inequalities.**    For all $a, b \in \mathbb{R}^d$ ($d \geq 1$) the following equality holds:

$$\|a\|^2 + \|b\|^2 = 2\langle a, b \rangle + \|a - b\|^2, \tag{10}$$

$$\langle a, b \rangle \leq \|a\| \cdot \|b\|. \tag{11}$$

**Coordinate-Lipschitz-smoothness.**    Throughout this paper, we assume that the smoothness condition (Assumption 1.1) is satisfied. This inequality can be represented in the equivalent form:

$$f(x + h\mathbf{e}_i) \leq f(x) + h\nabla_i f(x) + \frac{L_i h^2}{2}, \tag{12}$$

where $L_1, L_2, ..., L_d > 0$ for any $i \in [d], x \in \mathbb{R}^d$ and $h \in \mathbb{R}$.

**Lipschitz-smoothness.** To prove Theorem 4.1, we additionally assume $L_{[1-\alpha]}$-smoothness w.r.t. the norm $\|\cdot\|_{[1-\alpha]}$:

$$f(y) \leq f(x) + \langle \nabla f(x), y - x \rangle + \frac{L_{[1-\alpha]}}{2} \|y - x\|_{[1-\alpha]}^2, \quad \forall x, y \in \mathbb{R}^d. \tag{13}$$

### C.2 The Golden Ratio Method (GRM)

Algorithms 1 and 2, presented in Section 3 and 4, respectively, use the Golden Ratio Method (GRM) at least once per iteration. This method utilizes the oracle concept (2) considered in this paper and has the following form (See Algorithm 3).

---

**Algorithm 3** Golden Ratio Method (GRM)

---

1: **Input:** Interval $[a, b]$
2: **Initialization:** Choose constants $\epsilon > 0$ (desired accuracy), put the constant $\rho = \frac{1}{\Phi} = \frac{\sqrt{5}-1}{2}$
3: $\quad y \leftarrow a + (1 - \rho)(b - a)$
4: $\quad z \leftarrow a + \rho(b - a)$
5: **while** $b - a > \epsilon$ **do**
6: $\quad$ **if** $\phi(y, z) = -1$ **then**
7: $\quad\quad b \leftarrow z$
8: $\quad\quad z \leftarrow y$
9: $\quad\quad y \leftarrow a + (1 - \rho)(b - a)$
10: $\quad$ **else**
11: $\quad\quad a \leftarrow y$
12: $\quad\quad y \leftarrow z$
13: $\quad\quad z \leftarrow a + \rho(b - a)$
14: $\quad$ **end if**
15: **end while**
16: **Return:** $\frac{a+b}{2}$

---

We utilize the Golden Ratio Method to find a solution to the following one-dimensional problem:

$$\eta_k = \arg\min_{\eta \in \mathbb{R}} f(x_k + \eta \mathbf{e}_{i_k}).$$

Using the well-known fact about the golden ratio method that GRM is required to do $N = \mathcal{O}\left(\log \frac{1}{\epsilon}\right)$ (where $\epsilon$ is the accuracy of the solution to the linear search problem (by function)), we derive the following corollaries from the solution of this problem:

- In Section 4 (Accelerated Algorithms), for simplicity, we consider the scenario when the Order Oracle (2) is not subject to an adversarial noise ($\Delta = 0$) and the golden ratio method solves the inner problem exactly ($\epsilon \simeq 0$). Then we can observe the following:

$$f(x_k + \eta_k \mathbf{e}_{i_k}) \leq f(x_k + \eta \mathbf{e}_{i_k}), \qquad \forall \eta \in \mathbb{R}. \tag{14}$$

- In Section 3 ("Non-Accelerated Algorithms"), we consider the scenario when the Order Oracle (2) is subjected to an adversarial noise ($\Delta > 0$). Then from the convergence results ( by function) of the golden ratio method (GRM) we can observe the following:

$$f(x_k + \eta_k \mathbf{e}_{i_k}) \leq f(x_k + \eta \mathbf{e}_{i_k}) + \epsilon + c\Phi\Delta, \qquad \forall \eta \in \mathbb{R}, \tag{15}$$

where $c$ is some constant, $\epsilon$ is the GRM accuracy (by function) and $\Phi = \frac{1+\sqrt{5}}{2}$ is golden ratio.

It is worth noting that we consider convergence of the golden ratio in terms of function, because we assume that our Order Oracle may be subject to adversarial noise. If we talk about convergence by argument, there may be no convergence at all with a noisy concept of the Order Oracle. Thus, in the final corollary (15), we consider the scenario where adversarial noise accumulates over iterations, resulting in the following observation: the golden ratio method converges towards the $\mathcal{O}(\Delta)$ asymptote.

## D Proof of Convergence for Non-Accelerated Algorithm 1

In this section, we furnish the omitted proofs for the theorems presented in Section 3.

### D.1 Proof of Theorem 3.1

From Assumption 1.1 we obtain:

$$
\begin{aligned}
f\left(x_k + \eta_k \mathbf{e}_{i_k}\right) - f(x_k) &\overset{(15)}{\leq} f\left(x_k - \frac{1}{L_{i_k}} \nabla_{i_k} f(x_k) \mathbf{e}_{i_k}\right) - f(x_k) + \epsilon + c\Phi\Delta \\
&\overset{(12)}{\leq} -\frac{1}{L_{i_k}} (\nabla_{i_k} f(x_k))^2 + \frac{1}{2L_{i_k}} (\nabla_{i_k} f(x_k))^2 + \epsilon + c\Phi\Delta \\
&= -\frac{1}{2L_{i_k}} (\nabla_{i_k} f(x_k))^2 + \epsilon + c\Phi\Delta,
\end{aligned}
\tag{16}
$$

where $\eta_k = \arg\min_{\eta \in \mathbb{R}} f(x_k + \eta \mathbf{e}_{i_k})$. We use this as follows:

$$
\begin{aligned}
\mathbb{E}\left[f(x_{k+1})\right] - f(x_k) &= \mathbb{E}\left[f\left(x_k + \eta_k \mathbf{e}_{i_k}\right)\right] - f(x_k) \\
&\overset{(4)}{=} \sum_{i=1}^{d} p_\alpha(i) \left(f\left(x_k + \eta_k \mathbf{e}_{i_k}\right) - f(x_k)\right) \\
&\overset{(16)}{\leq} -\sum_{i=1}^{d} p_\alpha(i) \frac{1}{2L_i} (\nabla_i f(x_k))^2 + \sum_{i=1}^{d} p_\alpha(i) \left(\epsilon + c\Phi\Delta\right) \\
&= -\frac{1}{2S_\alpha} \left(\|\nabla f(x_k)\|_{[1-\alpha]}^*\right)^2 + \epsilon + c\Phi\Delta.
\end{aligned}
$$

Rearranging the terms and summing over all $k$, we have

$$
\begin{aligned}
\sum_{k=0}^{N-1} \frac{1}{2S_\alpha} \left(\|\nabla f(x_k)\|_{[1-\alpha]}^*\right)^2 &\leq \sum_{k=0}^{N-1} (F_k - F_{k+1}) + \sum_{k=0}^{N-1} \epsilon + \sum_{k=0}^{N-1} c\Phi\Delta \\
&\leq F_0 + F_N + N\epsilon + Nc\Phi\Delta \\
&\leq F_0 + N\epsilon + Nc\Phi\Delta,
\end{aligned}
$$

where $F_k = \mathbb{E}\left[f(x_k)\right] - f(x^*)$.

Dividing both sides by number of iterations $N$, we obtain the convergence rate for the non-convex case:

$$
\frac{1}{N} \sum_{k=0}^{N-1} \left(\|\nabla f(x_k)\|_{[1-\alpha]}^*\right)^2 \leq \frac{2S_\alpha}{N} F_0 + 2S_\alpha \epsilon + 2S_\alpha c\Phi\Delta.
$$

This convergence results imply the existence of a point $k \in [N]$ where holds true:

$$
\left(\|\nabla f(x_k)\|_{[1-\alpha]}^*\right)^2 \leq \mathcal{O}\left(\frac{S_\alpha F_0}{N} + S_\alpha \epsilon + S_\alpha \Phi\Delta\right).
$$

Then, achieving the desired accuracy $\varepsilon$, where $\|\nabla f(x_k)\|_{[1-\alpha]}^* \leq \varepsilon$, requires

$$
N = \mathcal{O}\left(\frac{S_\alpha F_0}{\varepsilon^2}\right), \qquad T = \tilde{\mathcal{O}}\left(\frac{S_\alpha F_0}{\varepsilon^2}\right)
$$

iterations and oracle calls respectively, provided the maximum noise does not exceed $\Delta \lesssim \varepsilon^2 / S_\alpha$.

### D.2 Proof of Theorem 3.2

From Assumption 1.1 we obtain:

$$
f\left(x_k + \eta_k \mathbf{e}_{i_k}\right) - f(x_k) \overset{(15)}{\leq} f\left(x_k - \frac{1}{L_{i_k}} \nabla_{i_k} f(x_k) \mathbf{e}_{i_k}\right) - f(x_k) + \epsilon + c\Phi\Delta
$$

$$\overset{(12)}{\leq} -\frac{1}{L_{i_k}}(\nabla_{i_k} f(x_k))^2 + \frac{1}{2L_{i_k}}(\nabla_{i_k} f(x_k))^2 + \epsilon + c\Phi\Delta$$

$$= -\frac{1}{2L_{i_k}}(\nabla_{i_k} f(x_k))^2 + \epsilon + c\Phi\Delta, \tag{17}$$

where $\eta_k = \arg\min_{\eta \in \mathbb{R}} f(x_k + \eta \mathbf{e}_{i_k})$. We use this as follows:

$$\mathbb{E}\left[f(x_{k+1})\right] - f(x_k) = \mathbb{E}\left[f\left(x_k + \eta_k \mathbf{e}_{i_k}\right)\right] - f(x_k)$$

$$\overset{(4)}{=} \sum_{i=1}^{d} p_\alpha(i) \left(f\left(x_k + \eta_k \mathbf{e}_{i_k}\right) - f(x_k)\right)$$

$$\overset{(17)}{\leq} -\sum_{i=1}^{d} p_\alpha(i)\frac{1}{2L_i}(\nabla_i f(x_k))^2 + \sum_{i=1}^{d} p_\alpha(i)\left(\epsilon + c\Phi\Delta\right)$$

$$= -\frac{1}{2S_\alpha}\left(\|\nabla f(x_k)\|_{[1-\alpha]}^*\right)^2 + \epsilon + c\Phi\Delta. \tag{18}$$

Denote $F_k = \mathbb{E}\left[f(x_k)\right] - f(x^*)$. Note that the above calculation can be used to show $f(x_{k+1}) \leq f(x_k)$, then we have

$$F_k \overset{\textcircled{1}}{\leq} \langle \nabla f(x_k), x_k - x^* \rangle$$

$$\overset{(11)}{\leq} \|x_k - x^*\|_{[1-\alpha]} \|\nabla f(x_k)\|_{[1-\alpha]}^*$$

$$\leq R_{[1-\alpha]} \|\nabla f(x_k)\|_{[1-\alpha]}^*, \tag{19}$$

where in $\textcircled{1}$ we used Assumption 1.2 with $\mu_{1-\alpha} = 0$, and a new notation for convenience, which looks as follows $R_{[1-\alpha]} = \sup_{x \in \mathbb{R}^d : f(x) \leq f(x_0)} \|x - x^*\|_{[1-\alpha]}$.

Then substituting (19) into (18) we obtain:

$$F_{k+1} \leq F_k - \frac{1}{2S_\alpha R_{[1-\alpha]}^2} F_k^2 + \epsilon + c\Phi\Delta.$$

Rewriting this inequality, we obtain:

$$\frac{1}{2S_\alpha R_{[1-\alpha]}^2} F_k^2 \leq F_k - F_{k+1} + \epsilon + c\Phi\Delta.$$

Next, we divide both sides by $F_{k+1}F_k$:

$$\frac{1}{2S_\alpha R_{[1-\alpha]}^2} \cdot \frac{F_k}{F_{k+1}} \leq \frac{1}{F_{k+1}} - \frac{1}{F_k} + \frac{\epsilon + c\Phi\Delta}{F_{k+1}F_k}.$$

Using the fact that $\frac{1}{2S_\alpha R_{[1-\alpha]}^2} \leq \frac{1}{2S_\alpha R_{[1-\alpha]}^2}\frac{F_k}{F_{k+1}}$ we obtain the following:

$$\frac{1}{2S_\alpha R_{[1-\alpha]}^2} \leq \frac{1}{F_{k+1}} - \frac{1}{F_k} + \frac{\epsilon + c\Phi\Delta}{F_{k+1}F_k}.$$

When summing over all $k$

$$\sum_{k=0}^{N-1} \frac{1}{2S_\alpha R_{[1-\alpha]}^2} \leq \sum_{k=0}^{N-1} \left(\frac{1}{F_{k+1}} - \frac{1}{F_k}\right) + \sum_{k=0}^{N-1} \frac{\epsilon + c\Phi\Delta}{F_{k+1}F_k},$$

we get:

$$N\frac{1}{2S_\alpha R_{[1-\alpha]}^2} \leq \frac{1}{F_N} - \frac{1}{F_0} + \frac{N\left(\epsilon + c\Phi\Delta\right)}{F_N F_{N-1}}$$

$$\leq \frac{1}{F_N} + \frac{N\left(\epsilon + c\Phi\Delta\right)}{F_N F_{N-1}}.$$

Taking into account the fact that $F_{N-1} = \mathbb{E}\left[f(x_{N-1})\right] - f(x^*) \geq \varepsilon$ and rewriting the expression we obtain the convergence rate for the convex case:

$$\mathbb{E}\left[f(x_N)\right] - f(x^*) \leq \frac{2S_\alpha R_{[1-\alpha]}^2}{N} + \frac{2S_\alpha R_{[1-\alpha]}^2}{\varepsilon}\left(\epsilon + c\Phi\Delta\right).$$

Then, achieving the desired accuracy $\varepsilon$, where $\mathbb{E}\left[f(x_N)\right] - f(x^*) \leq \varepsilon$, requires

$$N = \mathcal{O}\left(\frac{S_\alpha R_{[1-\alpha]}^2}{\varepsilon}\right), \qquad T = \tilde{\mathcal{O}}\left(\frac{S_\alpha R_{[1-\alpha]}^2}{\varepsilon}\right)$$

iterations and oracle calls respectively, provided the maximum noise does not exceed the following value $\Delta \lesssim \varepsilon^2/(S_\alpha R_{[1-\alpha]}^2)$.

### D.3 Proof of Theorem 3.3

From Assumption 1.1 we obtain:

$$
\begin{aligned}
f\left(x_k + \eta_k \mathbf{e}_{i_k}\right) - f(x_k) &\overset{(15)}{\leq} f\left(x_k - \frac{1}{L_{i_k}}\nabla_{i_k} f(x_k)\mathbf{e}_{i_k}\right) - f(x_k) + \epsilon + c\Phi\Delta \\
&\overset{(12)}{\leq} -\frac{1}{L_{i_k}}(\nabla_{i_k} f(x_k))^2 + \frac{1}{2L_{i_k}}(\nabla_{i_k} f(x_k))^2 + \epsilon + c\Phi\Delta \\
&= -\frac{1}{2L_{i_k}}(\nabla_{i_k} f(x_k))^2 + \epsilon + c\Phi\Delta, \qquad\qquad (20)
\end{aligned}
$$

where $\eta_k = \arg\min_{\eta \in \mathbb{R}} f(x_k + \eta \mathbf{e}_{i_k})$. We use this as follows:

$$
\begin{aligned}
\mathbb{E}\left[f(x_{k+1})\right] - f(x_k) &= \mathbb{E}\left[f\left(x_k + \eta_k \mathbf{e}_{i_k}\right)\right] - f(x_k) \\
&\overset{(4)}{=} \sum_{i=1}^d p_\alpha(i)\left(f\left(x_k + \eta_k \mathbf{e}_{i_k}\right) - f(x_k)\right) \\
&\overset{(20)}{\leq} -\sum_{i=1}^d p_\alpha(i)\frac{1}{2L_i}(\nabla_i f(x_k))^2 + \sum_{i=1}^d p_\alpha(i)\left(\epsilon + c\Phi\Delta\right) \\
&= -\frac{1}{2S_\alpha}\left(\|\nabla f(x_k)\|_{[1-\alpha]}^*\right)^2 + \epsilon + c\Phi\Delta. \qquad\qquad (21)
\end{aligned}
$$

By strong convexity, we have

$$
\begin{aligned}
f(x_k) - f(x^*) &\overset{\text{\textcircled{1}}}{\leq} \langle\nabla f(x_k), x_k - x^*\rangle - \frac{\mu_{1-\alpha}}{2}\|x_k - x^*\|_{[1-\alpha]} \\
&\overset{(11)}{\leq} \|\nabla f(x_k)\|_{[1-\alpha]}^* \cdot \|x_k - x^*\|_{[1-\alpha]} - \frac{\mu_{1-\alpha}}{2}\|x_k - x^*\|_{[1-\alpha]} \\
&\leq \frac{1}{\mu_{1-\alpha}}\left(\|\nabla f(x_k)\|_{[1-\alpha]}^*\right)^2,
\end{aligned}
$$

where in ① we used Assumption 1.2 with $\mu_{1-\alpha} > 0$. Then, using this inequality in (21) we have

$$\mathbb{E}\left[f(x_{k+1})\right] - f(x^*) \leq \left(1 - \frac{\mu_{1-\alpha}}{2S_\alpha}\right)\left(f(x_k) - f(x^*)\right) + \epsilon + c\Phi\Delta.$$

Applying recursion we obtain a linear convergence rate:

$$\mathbb{E}\left[f(x_N)\right] - f(x^*) \leq \left(1 - \frac{\mu_{1-\alpha}}{2S_\alpha}\right)^N\left(f(x_0) - f(x^*)\right) + \frac{2S_\alpha\epsilon}{\mu_{1-\alpha}} + \frac{2cS_\alpha\Phi\Delta}{\mu_{1-\alpha}}.$$

Then, achieving the desired accuracy $\varepsilon$, where $\mathbb{E}\left[f(x_N)\right] - f(x^*) \leq \varepsilon$, requires

$$N = \mathcal{O}\left(\frac{S_\alpha}{\mu_{1-\alpha}}\log\frac{1}{\varepsilon}\right), \qquad T = \tilde{\mathcal{O}}\left(\frac{S_\alpha}{\mu_{1-\alpha}}\log\frac{1}{\varepsilon}\right)$$

iterations and oracle calls respectively, provided the maximum noise does not exceed the following value $\Delta \lesssim \mu_{1-\alpha}\varepsilon/S_\alpha$.

# E  Proof of Convergence for Accelerated Algorithm 2

Denote $\omega_k = (1 - \beta_k)z_k + \beta_k y_k$. Then

$$y_k = \frac{(1 - \alpha_k)x_k}{1 - \alpha_k\beta_k} + \frac{\alpha_k(1 - \beta_k)}{1 - \alpha_k\beta_k} \cdot \frac{\omega_k - \beta_k y_k}{1 - \beta_k} = \frac{(1 - \alpha_k)x_k + \alpha_k\omega_k}{1 - \alpha_k\beta_k} - \frac{\alpha_k\beta_k y_k}{1 - \alpha_k\beta_k}.$$

Thus, in method we have the following representation:

$$y_k = (1 - \alpha_k)x_k + \alpha_k\omega_k. \tag{22}$$

Let the solution of initial problem denote $x^* = x_* = \arg\min_{x\in\mathbb{R}^d} f(x)$, then from Assumption 1.1 we obtain:

$$\frac{2L_{i_k}^\alpha a_{k+1}^2}{B_{k+1}^2 p_\beta^2(i_k)} \left( f(\underbrace{y_k + \eta_k \mathbf{e}_{i_k}}_{x_{k+1}}) - f(y_k) \right) \overset{(14)}{\leq} \frac{2L_{i_k}^\alpha a_{k+1}^2}{B_{k+1}^2 p_\beta^2(i_k)} \left( f\left( y_k - \frac{1}{L_{i_k}}\nabla_{i_k} f(y_k)\mathbf{e}_{i_k} \right) - f(y_k) \right)$$

$$\overset{(12)}{\leq} -\frac{a_{k+1}^2}{L_{i_k}^{1-\alpha} B_{k+1}^2 p_\beta^2(i_k)} \left( \nabla_{i_k} f(y_k) \right)^2 \pm \|\omega_k - x_*\|_{[1-\alpha]}^2$$

$$= -L_{i_k}^{1-\alpha} \left[ (\omega_k^{(i_k)} - x_*^{(i_k)})^2 + \left( \frac{a_{k+1}}{L_{i_k}^{1-\alpha} B_{k+1} p_\beta(i_k)}\nabla_{i_k} f(y_k) \right)^2 \right] - \sum_{i \neq i_k} L_i^{1-\alpha}(\omega_k^{(i)} - x_*^{(i)})^2$$

$$+ \|\omega_k - x_*\|_{[1-\alpha]}^2$$

$$\overset{(10)}{=} -L_{i_k}^{1-\alpha} \left[ \left( \omega_k^{(i_k)} - x_*^{(i_k)} - \frac{a_{k+1}}{L_{i_k}^{1-\alpha} B_{k+1} p_\beta(i_k)}\nabla_{i_k} f(y_k) \right)^2 + \frac{2a_{k+1}}{L_{i_k}^{1-\alpha} B_{k+1} p_\beta(i_k)} \left\langle \nabla_{i_k} f(y_k), \omega_k^{(i_k)} - x_*^{(i_k)} \right\rangle \right]$$

$$- \sum_{i \neq i_k} L_i^{1-\alpha}(\omega_k^{(i)} - x_*^{(i)})^2 + \|\omega_k - x_*\|_{[1-\alpha]}^2$$

$$= -\left\| \omega_k - \frac{a_{k+1}}{L_{i_k}^{1-\alpha} B_{k+1} p_\beta(i_k)}\nabla_{i_k} f(y_k)\mathbf{e}_{i_k} - x_* \right\|_{[1-\alpha]}^2 + \|\omega_k - x_*\|_{[1-\alpha]}^2$$

$$- \frac{2a_{k+1}}{B_{k+1} p_\beta(i_k)} \left\langle \nabla_{i_k} f(y_k), \omega_k^{(i_k)} - x_*^{(i_k)} \right\rangle + \frac{2}{\mu_{1-\alpha}} \left\langle \nabla f(x_*), z_{k+1} - x_* \right\rangle$$

$$\overset{①}{\leq} -\left\| \omega_k - \frac{a_{k+1}}{L_{i_k}^{1-\alpha} B_{k+1} p_\beta(i_k)}\nabla_{i_k} f(y_k)\mathbf{e}_{i_k} - x_* \right\|_{[1-\alpha]}^2 + \|\omega_k - x_*\|_{[1-\alpha]}^2$$

$$- \frac{2a_{k+1}}{B_{k+1} p_\beta(i_k)} \left\langle \nabla_{i_k} f(y_k), \omega_k^{(i_k)} - x_*^{(i_k)} \right\rangle - \|z_{k+1} - x_*\|_{[1-\alpha]}^2 + \frac{2}{\mu_{1-\alpha}} \left( f(z_{k+1}) - f(x_*) \right)$$

$$\overset{(13)}{\leq} -\|z_{k+1} - x_*\|_{[1-\alpha]}^2 + \|\omega_k - x_*\|_{[1-\alpha]}^2 - \frac{2a_{k+1}}{B_{k+1} p_\beta(i_k)} \left\langle \nabla_{i_k} f(y_k), \omega_k^{(i_k)} - x_*^{(i_k)} \right\rangle$$

$$+ \frac{2}{\mu_{1-\alpha}} \left[ f(z_{k+1}) - f(x_*) \right] + \frac{2}{L_{1-\alpha}} \left[ f(x_*) - f\left( \omega_k - \frac{a_{k+1}}{L_{i_k}^{1-\alpha} B_{k+1} p_\beta(i_k)}\nabla_{i_k} f(y_k)\mathbf{e}_{i_k} \right) \right]$$

$$- \frac{2}{L_{1-\alpha}} \left\langle \nabla f(x_*), \omega_k - \frac{a_{k+1}}{L_{i_k}^{1-\alpha} B_{k+1} p_\beta(i_k)}\nabla_{i_k} f(y_k)\mathbf{e}_{i_k} - x_* \right\rangle$$

$$\leq -\|z_{k+1} - x_*\|_{[1-\alpha]}^2 + \|\omega_k - x_*\|_{[1-\alpha]}^2 - \frac{2a_{k+1}}{B_{k+1} p_\beta(i_k)} \left\langle \nabla_{i_k} f(y_k), \omega_k^{(i_k)} - x_*^{(i_k)} \right\rangle$$

$$+ \frac{2}{\sigma_{1-\alpha}} \left[ f(z_{k+1}) - f(x_*) + f(x_*) - f\left( \omega_k - \frac{a_{k+1}}{L_{i_k}^{1-\alpha} B_{k+1} p_\beta(i_k)}\nabla_{i_k} f(y_k)\mathbf{e}_{i_k} \right) \right]$$

$$\overset{(14)}{\leq} -\|z_{k+1} - x_*\|_{[1-\alpha]}^2 + \|\omega_k - x_*\|_{[1-\alpha]}^2 - \frac{2a_{k+1}}{B_{k+1} p_\beta(i_k)} \left\langle \nabla_{i_k} f(y_k), \omega_k^{(i_k)} - x_*^{(i_k)} \right\rangle, \tag{23}$$

where in ① we used Assumption 1.2 with $\mu_{1-\alpha} > 0$.

Denote $r_k^2 = \|z_k - x_*\|_{[1-\alpha]}^2$ and $\omega_k = (1 - \beta_k)z_k + \beta_k y_k$, then due to convexity of the norm function it follows:

$$\|\omega_k - x_*\|_{[1-\alpha]}^2 \le (1 - \beta_k) \|z_k - x_*\|_{[1-\alpha]}^2 + \beta_k \|y_k - x_*\|_{[1-\alpha]}^2 . \tag{24}$$

Substituting (24) into (23) we obtain

$$
\begin{aligned}
B_{k+1}r_{k+1}^2 &\overset{(24)}{\le} (1 - \beta_k)B_{k+1}r_k^2 + \beta_k B_{k+1}\|y_k - x_*\|_{[1-\alpha]}^2 \\
&\quad - \frac{2a_{k+1}}{p_\beta(i_k)}\left\langle \nabla_{i_k} f(y_k)\mathbf{e}_{i_k}, \omega_k^{i_k} - x_*^{i_k}\right\rangle + \frac{2L_{i_k}^\alpha a_{k+1}^2}{B_{k+1}p_\beta^2(i_k)}\left(f(y_k) - f(x_{k+1})\right) \\
&\overset{①}{\le} B_k r_k^2 + \beta_k B_{k+1}\|y_k - x_*\|_{[1-\alpha]}^2 - \frac{2a_{k+1}}{p_\beta(i_k)}\left\langle \nabla_{i_k} f(y_k)\mathbf{e}_{i_k}, \omega_k^{i_k} - x_*^{i_k}\right\rangle + \frac{2L_{i_k}^\alpha a_{k+1}^2}{B_{k+1}p_\beta^2(i_k)}\left(f(y_k) - f(x_{k+1})\right) \\
&\overset{(4)}{\le} B_k r_k^2 + \beta_k B_{k+1}\|y_k - x_*\|_{[1-\alpha]}^2 - \frac{2a_{k+1}}{p_\beta(i_k)}\left\langle \nabla_{i_k} f(y_k)\mathbf{e}_{i_k}, \omega_k^{i_k} - x_*^{i_k}\right\rangle + \frac{2L_{i_k}^\alpha a_{k+1}^2 S_\beta^2}{B_{k+1}L_{i_k}^{2\beta}}\left(f(y_k) - f(x_{k+1})\right) \\
&\overset{②}{=} B_k r_k^2 + \beta_k B_{k+1}\|y_k - x_*\|_{[1-\alpha]}^2 - \frac{2a_{k+1}}{p_\beta(i_k)}\left\langle \nabla_{i_k} f(y_k)\mathbf{e}_{i_k}, \omega_k^{i_k} - x_*^{i_k}\right\rangle + \frac{2a_{k+1}^2 S_\beta^2}{B_{k+1}}\left(f(y_k) - f(x_{k+1})\right),
\end{aligned}
$$

where in ① we use that $(1 - \beta_k)B_{k+1} = B_{k+1} - \sigma_{1-\alpha}a_{k+1} = B_k$, and in ② we use that $\alpha = 2\beta$.

Note that $\mathbb{E}\left[f(x_{k+1})\right] = \sum_{i=1}^d p_\beta(i)f(x_{k+1})$. Therefore, taking expectation we obtain:

$$
\begin{aligned}
\mathbb{E}\left[B_{k+1}r_{k+1}^2\right] &\le B_k r_k^2 + \beta_k B_{k+1}\|y_k - x_*\|_{[1-\alpha]}^2 - \mathbb{E}\left[\frac{2a_{k+1}}{p_\beta(i_k)}\left\langle \nabla_{i_k} f(y_k)\mathbf{e}_{i_k}, \omega_k^{i_k} - x_*^{i_k}\right\rangle\right] \\
&\quad + \frac{2a_{k+1}^2 S_\beta^2}{B_{k+1}}\left(f(y_k) - \mathbb{E}\left[f(x_{k+1})\right]\right) \\
&\le B_k r_k^2 + \beta_k B_{k+1}\|y_k - x_*\|_{[1-\alpha]}^2 - \sum_{i=1}^d p_\beta(i)\frac{2a_{k+1}}{p_\beta(i)}\left\langle \nabla_i f(y_k)\mathbf{e}_i, \omega_k^i - x_*^i\right\rangle \\
&\quad + \frac{2a_{k+1}^2 S_\beta^2}{B_{k+1}}\left(f(y_k) - \mathbb{E}\left[f(x_{k+1})\right]\right) \\
&= B_k r_k^2 + \beta_k B_{k+1}\|y_k - x_*\|_{[1-\alpha]}^2 + 2a_{k+1}\left\langle \nabla f(y_k), x_* - \omega_k\right\rangle + \frac{2a_{k+1}^2 S_\beta^2}{B_{k+1}}\left(f(y_k) - \mathbb{E}\left[f(x_{k+1})\right]\right).
\end{aligned}
$$

Since $\omega_k \overset{(22)}{=} y_k + \frac{1-\alpha_k}{\alpha_k}(y_k - x_k)$, we obtain

$$
\begin{aligned}
2a_{k+1}\left\langle \nabla f(y_k), x_* - \omega_k\right\rangle &= 2a_{k+1}\left\langle \nabla f(y_k), x_* - y_k + \frac{1-\alpha_k}{\alpha_k}(x_k - y_k)\right\rangle \\
&\overset{①}{\le} 2a_{k+1}\left(f(x_*) - f(y_k)\right) - a_{k+1}\sigma_{1-\alpha}\|x_* - y_k\|_{[1-\alpha]}^2 \\
&\quad + 2a_{k+1}\frac{1-\alpha_k}{\alpha_k}\left(f(x_k) - f(y_k)\right) \\
&\overset{②}{=} 2a_{k+1}f(x_*) - 2A_{k+1}f(y_k) + 2A_k f(x_k) - a_{k+1}\sigma_{1-\alpha}\|x_* - y_k\|_{[1-\alpha]}^2, \tag{25}
\end{aligned}
$$

where in ① we use Assumption 1.2 with $\mu_{1-\alpha} > 0$ and in ② we use that $a_{k+1}\frac{1-\alpha_k}{\alpha_k} = a_{k+1}\frac{1 - \frac{a_{k+1}}{A_{k+1}}}{\frac{a_{k+1}}{A_{k+1}}} = A_{k+1} - a_{k+1} = A_k$.

$$\mathbb{E}\left[B_{k+1}r_{k+1}^2\right] \le B_k r_k^2 + \beta_k B_{k+1}\|y_k - x_*\|_{[1-\alpha]}^2 + 2a_{k+1}\left\langle \nabla f(y_k), x_* - \omega_k\right\rangle + \frac{2a_{k+1}^2 S_\beta^2}{B_{k+1}}\left(f(y_k) - \mathbb{E}\left[f(x_{k+1})\right]\right)$$

$$\stackrel{①}{=} B_k r_k^2 + a_{k+1}\sigma_{1-\alpha} \|y_k - x_*\|_{[1-\alpha]}^2 + 2a_{k+1}\langle \nabla f(y_k), x_* - \omega_k\rangle + \frac{2a_{k+1}^2 S_\beta^2}{B_{k+1}}\left(f(y_k) - \mathbb{E}\left[f(x_{k+1})\right]\right)$$

$$\stackrel{(25)}{\leq} B_k r_k^2 + 2a_{k+1}f(x_*) - 2A_{k+1}f(y_k) + 2A_k f(x_k) \pm 2A_k f(x_*) + \frac{2a_{k+1}^2 S_\beta^2}{B_{k+1}}\left(f(y_k) - \mathbb{E}\left[f(x_{k+1})\right]\right)$$

$$\stackrel{②}{=} B_k r_k^2 + 2a_{k+1}f(x_*) - 2A_{k+1}f(y_k) + 2A_k f(x_k) \pm 2A_k f(x_*) + 2A_{k+1}\left(f(y_k) - \mathbb{E}\left[f(x_{k+1})\right]\right)$$

$$= B_k r_k^2 - 2A_{k+1}\left(\mathbb{E}\left[f(x_{k+1})\right] - f(x_*)\right) + 2A_k\left(f(x_k) - f(x_*)\right),$$

where in ① we use that $\beta_k = \frac{\sigma_{1-\alpha}a_{k+1}}{B_{l+1}}$, and in ② we use that $a_{k+1}^2 S_\beta^2 = A_{k+1}B_{k+1}$.

By summing over $k$ we obtain:

$$2\sum_{k=0}^{N-1} A_{k+1}\left(\mathbb{E}\left[f(x_{k+1})\right] - f(x_*)\right) \leq 2\sum_{k=0}^{N-1} A_k\left(f(x_k) - f(x_*)\right) + \sum_{k=0}^{N-1} B_k r_k^2 - \sum_{k=0}^{N-1} \mathbb{E}\left[B_{k+1}r_{k+1}^2\right].$$

$$2A_N\left(\mathbb{E}\left[f(x_N)\right] - f(x_*)\right) \leq 2A_0\left(f(x_0) - f(x_*)\right) + B_0 r_0^2 - \mathbb{E}\left[B_N r_N^2\right].$$

Using the known facts from Nesterov and Stich (2017) we can estimate the parameters $A_N$ and $B_N$:

$$A_N \geq \frac{1}{4\mu_{1-\alpha}}\left[(1+\gamma)^N - (1-\gamma)^N\right]^2$$

$$\geq \frac{1}{4\mu_{1-\alpha}}\left[(1+\gamma)^N - 1\right]^2 \geq \frac{1}{8\mu_{1-\alpha}}(1+\gamma)^{2N} \geq \frac{1}{8\mu_{1-\alpha}}(1+\gamma)^N,$$

$$B_N \geq \frac{1}{4}\left[(1+\gamma)^N + (1-\gamma)^N\right]^2,$$

where $\gamma = \frac{\sqrt{\mu_{1-\alpha}}}{2S_{\alpha/2}}$. Then using $A_0 = 0$ and $B_0 = 1$ we have

$$A_N\left(\mathbb{E}\left[f(x_N)\right] - f(x_*)\right) \leq 2A_0\left(f(x_0) - f(x_*)\right) + B_0 r_0^2 - B_N \mathbb{E}\left[r_N^2\right].$$

$$\stackrel{①}{\leq} \frac{1}{\mu_{1-\alpha}}\left(f(x_0) - f(x_*) - \langle\nabla f(x_*), x_0 - x_*\rangle\right)$$

$$= \frac{1}{\mu_{1-\alpha}}\left(f(x_0) - f(x_*)\right).$$

Let's divide both parts by $A_N$, then we have the convergence rate for accelerated method

$$\mathbb{E}\left[f(x_N)\right] - f(x_*) \leq \frac{1}{\mu_{1-\alpha}A_N}\left(f(x_0) - f(x_*)\right)$$

$$\leq 8(1+\gamma)^{-N}\left(f(x_0) - f(x_*)\right)$$

$$\leq 8(1-\gamma)^N\left(f(x_0) - f(x_*)\right)$$

$$= 8\left(1 - \frac{\sqrt{\mu_{1-\alpha}}}{2S_{\alpha/2}}\right)^N\left(f(x_0) - f(x_*)\right).$$

Then, achieving the desired accuracy $\varepsilon$, where $\mathbb{E}\left[f(x_N)\right] - f(x^*) \leq \varepsilon$, requires

$$N = \mathcal{O}\left(\frac{S_{\alpha/2}}{\sqrt{\mu_{1-\alpha}}}\log\frac{1}{\varepsilon}\right), \qquad T = \tilde{\mathcal{O}}\left(\frac{S_{\alpha/2}}{\sqrt{\mu_{1-\alpha}}}\log\frac{1}{\varepsilon}\right)$$

iterations and oracle calls respectively.

# F   Scheme of the Proof of Asymptotic Convergence of the Algorithm with the Stochastic Order Oracle Concept

In this Section, we give a proof of Theorem 5.3, which is based on the work of Polyak and Tsypkin (1980). In order to take advantage of the above work, we need to show that our method (9) is a normalized stochastic gradient descent. Therefore, we will first prove two auxiliary Lemmas 5.1 and 5.2 showing that our algorithm is still a normalized stochastic gradient descent, and then proceed to the statement of the Theorem 5.3. Our reasoning in the Proofs of auxiliary Lemmas is similar to the work of Saha et al. (2021).

**Lemma F.1.** *Let the function $f$ be L-smooth (for all $x, y \in \mathbb{R}^d$ it holds: $f(y, \xi) \le f(x, \xi) + \langle \nabla f(x, \xi), y - x \rangle + \frac{L}{2} \|y - x\|^2$), $\gamma = \frac{\|\nabla f(x, \xi)\|}{\sqrt{dL}}$ and $\mathbf{e} \in S^d(1)$, then the following holds:*

$$\phi(x + \gamma \mathbf{e}, x - \gamma \mathbf{e}, \xi) \mathbf{e} = \operatorname{sign}\left[\langle \nabla f(x, \xi), \mathbf{e} \rangle\right] \mathbf{e}.$$

*Proof.* From the Assumption of $L$-smoothness of the function $f$ we have:

$$\langle \nabla f(x, \xi), \gamma \mathbf{e} \rangle - \frac{L\gamma^2}{2} \le f(x + \gamma \mathbf{e}, \xi) - f(x, \xi) \le \langle \nabla f(x, \xi), \gamma \mathbf{e} \rangle + \frac{L\gamma^2}{2};$$

$$-\langle \nabla f(x, \xi), \gamma \mathbf{e} \rangle - \frac{L\gamma^2}{2} \le f(x - \gamma \mathbf{e}, \xi) - f(x, \xi) \le -\langle \nabla f(x, \xi), \gamma \mathbf{e} \rangle + \frac{L\gamma^2}{2}.$$

Subtracting the inequalities, we obtain

$$|f(x + \gamma \mathbf{e}, \xi) - f(x - \gamma \mathbf{e}, \xi) - 2\gamma \langle \nabla f(x, \xi), \mathbf{e} \rangle| \le L\gamma^2.$$

Consequently, if $L\gamma^2 \le \gamma |\langle \nabla f(x, \xi), \mathbf{e} \rangle|$, we have that

$$\phi(x + \gamma \mathbf{e}, x - \gamma \mathbf{e}, \xi) \mathbf{e} = \operatorname{sign}\left[\langle \nabla f(x, \xi), \mathbf{e} \rangle\right] \mathbf{e}.$$

Let's analyse $\mathcal{P}_\mathbf{e}\left(L\gamma \le |\langle \nabla f(x, \xi), \mathbf{e} \rangle|\right)$. It is known that for $\mathbf{u} \sim \mathcal{N}(0, I)$, $\mathbf{e} := \frac{\mathbf{u}}{\|\mathbf{u}\|}$ is a vector uniformly distributed on the unit Euclidean sphere. Then the following is true:

$$\mathcal{P}_\mathbf{e}\left(|\langle \nabla f(x, \xi), \mathbf{e} \rangle| \ge L\gamma\right) = \mathcal{P}_\mathbf{u}\left(|\langle \nabla f(x, \xi), \mathbf{u} \rangle| \ge L\gamma \|\mathbf{u}\|\right)$$

$$\le \mathcal{P}_\mathbf{u}\left(|\langle \nabla f(x, \xi), \mathbf{u} \rangle| \ge 2L\gamma\sqrt{d \log 1/\tilde{\beta}}\right) + \mathcal{P}_\mathbf{u}\left(\|\mathbf{u}\| \ge 2\sqrt{d \log 1/\tilde{\beta}}\right)$$

$$\le \mathcal{P}_\mathbf{u}\left(|\langle \nabla f(x, \xi), \mathbf{u} \rangle| \ge 2L\gamma\sqrt{d \log 1/\tilde{\beta}}\right) + \tilde{\beta},$$

where we used the well-known fact that $\forall \tilde{\beta}$: $\mathcal{P}_\mathbf{u}\left(\|\mathbf{u}\| \le 2\sqrt{d \log 1/\tilde{\beta}}\right) \ge 1 - \tilde{\beta}$. On the other hand, since $\langle \nabla f(x, \xi), \mathbf{u} \rangle \sim \mathcal{N}\left(0, \|\nabla f(x, \xi)\|^2\right)$, then for all $\gamma > 0$ we obtain

$$\mathcal{P}\left(|\langle \nabla f(x, \xi), \mathbf{u} \rangle| \le \gamma\right) \le \frac{2\gamma}{\|\nabla f(x, \xi)\| \sqrt{2\pi}} \le \frac{\gamma}{\|\nabla f(x, \xi)\|}.$$

Combining the inequalities, we have that $\phi(x + \gamma \mathbf{e}, x - \gamma \mathbf{e}, \xi) \mathbf{e} = \operatorname{sign}\left[\langle \nabla f(x, \xi), \mathbf{e} \rangle\right] \mathbf{e}$ except with probability at most

$$\inf_{\tilde{\beta} > 0} \left\{\tilde{\beta} + \frac{2L\gamma\sqrt{d \log 1/\tilde{\beta}}}{\|\nabla f(x, \xi)\|}\right\} \le \frac{3L\gamma}{\|\nabla f(x, \xi)\|} \sqrt{d \log \frac{\|\nabla f(x, \xi)\|}{\sqrt{dL}\gamma}} \overset{\text{①}}{=} 0 = \beta,$$

where in ① we take $\gamma = \frac{\|\nabla f(x, \xi)\|}{\sqrt{dL}}$. Thus, given the condition on the smoothing parameter, the statement of the Lemma is satisfied with probability 1.

$\square$

**Lemma F.2.** *Let vector $\nabla f(x, \xi) \in \mathbb{R}^d$ and vector $\mathbf{e} \in S^d(1)$, then we have*

$$\mathbb{E}\left[\operatorname{sign}\left[\langle \nabla f(x, \xi), \mathbf{e} \rangle\right] \mathbf{e}\right] = \frac{c}{\sqrt{d}} \cdot \frac{\nabla f(x, \xi)}{\|\nabla f(x, \xi)\|},$$

*where $c \in [\frac{1}{20}, 1]$ is some universal constant.*

*Proof.* Without loss of generality, we can assume $\|\nabla f(x, \xi)\| = 1$, as normalizing $\|\nabla f(x, \xi)\|$ does not impact the left-hand side. First, let's demonstrate that $\mathbb{E}\left[\operatorname{sign}\left[\langle \nabla f(x, \xi), \mathbf{e} \rangle\right] \mathbf{e}\right] = \zeta \nabla f(x, \xi)$ for some $\zeta \in \mathbb{R}$. Consider the reflection matrix along $\nabla f(x, \xi)$ given by:

$$P = 2\nabla f(x, \xi)\nabla f(x, \xi)^\mathrm{T} - I,$$

and examine the random vector $\tilde{\mathbf{e}} = P\mathbf{e}$. As you can see that

$$\text{sign}\left[\langle \nabla f(x,\xi), \tilde{\mathbf{e}}\rangle\right] = \text{sign}\left[2\|\nabla f(x,\xi)\|^2 \langle \nabla f(x,\xi), \mathbf{e}\rangle - \langle \nabla f(x,\xi), \mathbf{e}\rangle\right] = \text{sign}\left[\langle \nabla f(x,\xi), \mathbf{e}\rangle\right].$$

Since $\tilde{\mathbf{e}}$ is also a random vector on the unit sphere, we then have

$$\begin{aligned}
\mathbb{E}_{\mathbf{e}}\left[\text{sign}\left[\langle \nabla f(x,\xi), \mathbf{e}\rangle\right]\mathbf{e}\right] &= \frac{1}{2}\mathbb{E}_{\mathbf{e}}\left[\text{sign}\left[\langle \nabla f(x,\xi), \mathbf{e}\rangle\right]\mathbf{e}\right] + \frac{1}{2}\mathbb{E}_{\mathbf{e}}\left[\text{sign}\left[\langle \nabla f(x,\xi), \tilde{\mathbf{e}}\rangle\right]\tilde{\mathbf{e}}\right] \\
&= \frac{1}{2}\mathbb{E}_{\mathbf{e}}\left[\text{sign}\left[\langle \nabla f(x,\xi), \mathbf{e}\rangle\right]\mathbf{e}\right] \\
&\qquad + \frac{1}{2}\mathbb{E}_{\mathbf{e}}\left[\text{sign}\left[\langle \nabla f(x,\xi), \mathbf{e}\rangle\right]\left(2\nabla f(x,\xi)\nabla f(x,\xi)^{\mathsf{T}} - I\right)\mathbf{e}\right] \\
&= \mathbb{E}_{\mathbf{e}}\left[\langle \nabla f(x,\xi), \mathbf{e}\rangle \text{sign}\left[\langle \nabla f(x,\xi), \mathbf{e}\rangle\right]\right]\nabla f(x,\xi).
\end{aligned}$$

Thus, $\mathbb{E}_{\mathbf{e}}\left[\text{sign}\left[\langle \nabla f(x,\xi), \mathbf{e}\rangle\right]\mathbf{e}\right] = \zeta \nabla f(x)$, where $\zeta = \mathbb{E}_{\mathbf{e}}\left[|\langle \nabla f(x,\xi), \mathbf{e}\rangle|\right]$. It remains to restrict $\zeta$, which by virtue of rotation invariance is equal to $\zeta = \mathbb{E}\left[e_1\right]$. For an upper bound, observe that by symmetry $\mathbb{E}\left[e_1^2\right] = \frac{1}{d}\mathbb{E}\left[\sum_{i=1}^{d} e_i^2\right] = \frac{1}{d}$ and consequently:

$$\mathbb{E}\left[|e_1|\right] \le \sqrt{\mathbb{E}\left[e_1^2\right]} = \frac{1}{\sqrt{d}}.$$

Let us prove a lower bound on $\zeta$. If $\mathbf{e}$ were a Gaussian random vector with i.i.d. entries $e_i \sim \mathcal{N}\left(0, \frac{1}{d}\right)$, then from the standard properties of the (truncated) Gaussian distribution we would obtain that $\mathbb{E}\left[|e_1|\right] = \sqrt{\frac{2}{\pi d}}$. For $\mathbf{e}$ uniformly distributed on the unit sphere, $e_i$ is distributed like $\frac{u_1}{\|\mathbf{u}\|}$, where $\mathbf{u}$ is Gaussian with i.i.d. entries $\mathcal{N}\left(0, \frac{1}{d}\right)$. We then can write:

$$\begin{aligned}
\mathcal{P}\left(|e_1| \ge \frac{v}{\sqrt{d}}\right) &= \mathcal{P}\left(\frac{|u_1|}{\|\mathbf{u}\|} \ge \frac{v}{\sqrt{d}}\right) \ge \mathcal{P}\left(|u_1| \ge \frac{1}{\sqrt{d}} \text{ and } \|\mathbf{u}\| \le \frac{1}{v}\right) \\
&\ge 1 - \mathcal{P}\left(|u_1| \ge \frac{1}{\sqrt{d}}\right) - \mathcal{P}\left(\|\mathbf{u}\| > \frac{1}{v}\right).
\end{aligned}$$

Since $u_1\sqrt{d}$ is a standard Normal, we obtain

$$\mathcal{P}\left(|u_1| \ge \frac{1}{\sqrt{d}}\right) = \mathcal{P}\left(-1 < u_1\sqrt{d} < 1\right) \le 0.7,$$

and since $\mathbb{E}\left[\|\mathbf{u}\|^2\right] = 1$, the application of Markov inequality gives:

$$\mathcal{P}\left(\|\mathbf{u}\| > \frac{1}{v}\right) = \mathcal{P}\left(\|\mathbf{u}\|^2 > \frac{1}{v^2}\right) \le v^2 \mathbb{E}\left[\|\mathbf{u}\|^2\right] = v^2.$$

For $v = 0.25$, this means that $\mathcal{P}\left(|e_1| \ge 0.25\sqrt{d}\right) \ge 0.2$, whence $\zeta = \mathbb{E}\left[|e_1|\right] \ge \frac{1}{20}\sqrt{d}$.

$\square$

Now that we have shown that our method (9) is a normalized stochastic gradient descent, then by applying Theorem 2 of Polyak and Tsypkin (1980), and refining Theorem 2 to the case of normalized SGD, where in our case

$$R(x) = \mathbb{E}\left[\phi(x + \gamma\mathbf{e}, x - \gamma\mathbf{e}, \xi)\mathbf{e}\right] = \mathbb{E}\left[\text{sign}\left[\langle \nabla f(x,\xi), \mathbf{e}\rangle\right]\mathbf{e}\right] \overset{①}{=} \frac{c}{\sqrt{d}}\mathbb{E}_{\xi}\left[\frac{\nabla f(x,\xi)}{\|\nabla f(x,\xi)\|}\right]$$

$$\overset{②}{=} \frac{c}{\sqrt{d}}\int \varphi(\nabla f(x) + z)dP(z) = \frac{c}{\sqrt{d}}\psi(\nabla f(x)),$$

where in ① we used the Lemma 5.2, and in ② we defined a function $\varphi(z) = \frac{z}{\|z\|}$ and used that $\nabla f(x,\xi) = \nabla f(x) + \xi$. Thus, using the following

$$R'(x^*) = \frac{c}{\sqrt{d}}\psi'(0)\nabla^2 f(x^*) = \frac{c}{\sqrt{d}}\int \nabla\varphi(z)dP(z)\nabla^2 f(x^*) = \left(1 - \frac{1}{d}\right)\frac{c}{\sqrt{d}}\underbrace{\int \|z\|^{-1}dP(z)}_{\alpha}\nabla^2 f(x^*)$$

and $\eta_k = \frac{\eta}{k}$, we obtain the sought statement of Theorem 5.3, which guarantees asymptotic convergence.

# G  Description of the "Private Communication" Approach

In this section, we describe an approach privately communicated to us by Yurii Nesterov to create a more efficient algorithm for *low-dimensional* problems, compared to the gradient-based methods proposed in this paper, using only the Order Oracle (2).

Consider the following method for solving the problem of minimizing a convex Lipschitz function (with Lipschitz constant $L$) on a square in $\mathbb{R}^2$ with side $R$. A *horizontal line* (blue line) is drawn through the center of the square. On the interval carved from the square of this line, with accuracy $\sim \varepsilon / \log(LR/\varepsilon)$ (by function) we solve the one-dimensional optimization (line search) problem. At the found point (blue point), the direction of the function gradient (which is determined via the Order Oracle) is calculated and it is determined in which of the two rectangles it "looks"; this rectangle is "discarded". A *vertical line* (red line) is drawn through the center of the remaining rectangle, and on the segment carved by this line in the rectangle, also with accuracy $\sim \varepsilon / \log(LR/\varepsilon)$ (by function) solves the problem of one-dimensional optimization (line search) problem. At the found point (red point), the direction of the function gradient (which is determined via the Order Oracle) is calculated and it is determined in which of the two rectangles it "looks"; this rectangle is "discarded". As a result of this procedure, the linear size of the original square is halved (green square). The schematic of the performance of the algorithm is shown in Figure 7.

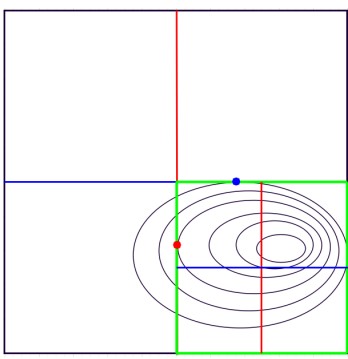

Figure 7: Schematic of algorithm

**Corollary G.1.** *It is not difficult to show that after $\log(LR/\varepsilon)$ repetitions of such a procedure we can find with $\varepsilon$ accuracy (by function) the solution of the original problem.*

