# OpenReview forum: "Acceleration Exists! Optimization Problems When Oracle Can Only Compare Objective Function Values"
_NeurIPS.cc/2024/Conference — NeurIPS 2024 poster_

### Official Review · Reviewer_ErqJ · 2024-07-03

**Soundness:** 3
**Presentation:** 3
**Contribution:** 2
**Rating:** 5
**Confidence:** 3

**Summary:**

In this paper the authors consider the problem of $\min_x f(x)$ with an order oracle that returns $sign (f(x)-f(y)+\delta)$ for some bounded noise $\delta$. The method is based on line search integrated with existing randomized coordinate update algorithms. Convergence rates on non-convex, convex, strongly convex objectives and its accelerated variant in the strongly convex case are provided. These are further extended to the stochastic setup $\min_x E_\zeta[f_\zeta (x)]$ with oracle $sign(f_\zeta(x)-f_\zeta(y))$ where a connection to normalized SGD is drawn and asymptotic convergence is derived. Numerical experiments are presented illustrating the performance of the proposed algorithms.

**Strengths:**

The paper has clear exposition and adequately surveys the related literature. The problem under study is of great interest to the machine learning community and has many interesting applications. The technical claims are sound and presented ideas are conceptually simple with demonstrated good performance in several setting.

**Weaknesses:**

I don't think the analysis result of "the convergence rate of random coordinate descent with the order oracle (OrderRCD), assuming minimal noise, is equal to first-order method random coordinate descent (RCD)" is surprising, given the algorithm is searching for best (1D) stepsize for coordinate update. And in this regard, neither is the acceleration result based on existing algorithmic framework. I've listed my main questions in the section below and here are some minor comments:

- Line 77 is not finished. Line 78: it would be more clear if one could put in the argument in the norm instead of $\|\cdot\|$, which presumably would be on $x$?

- It would be good to put the GRM in the main text otherwise Algorithm 1 does not make it clear how order oracle is used, which is the main focus of the paper.

- Line 179: I'd suggest using a different letter for line search accuracy instead of $\epsilon$, which was used for denoting desired precision of the final optimization problem elsewhere.

- In (8), it would be better to use the subscript $f_\zeta (x)$, which would make the notation more consistent with (1).

**Questions:**

- Why is coordinate descent the best thing to do? To me, it seems like the order oracle is essentially a "noisy normalized gradient oracle". So while I can understand it is one way to incorporate the oracle information into existing method, is there any sense in which this is the optimal way to use this information? I think the paper will benefit from such a justification otherwise it seems rather ad-hoc. For example, why is coordinate update a good idea in this setup? Most zeroth-order methods do not seem to rely heavily on this (note that the oracle information is only used to select the stepsize in Algorithm 1).

- Algorithm 1 as stated works on deterministic objective, i.e., it is not exploiting any finite-sum-type structure. If parts of the result do not concern stochastic objective, the authors should redefine their problem statement in (1).

- The accelerated algorithm 2 has several loops, which makes me wonder how well it would be able to deal with noise in the oracle.

- Similarly in Section 5, these reductions to existing methods are useful for getting some convergence guarantees, but is there a way to see this is how one should use the information optimally?

**Limitations:**

Yes it's adequately addressed.

---

> ### Author Rebuttal · Authors · 2024-08-04
>
> Dear **Reviewer ErqJ**,
>
> We thank you for your feedback on our work.
>
> > **I don't think the analysis result of...**
>
> We provide detailed answers to all questions below, including concerns about the significance and novelty of our results.
>
> > **Line 77 is not finished. Line 78...**
>
> Thank you, we will correct those typos.
>
> > **It would be good to put the GRM...**
>
> Due to space limitations in the main part of the paper, we had to move the GRM algorithm (which is not small) to Appendix. However, we have provided all the important information needed to understand our results in the main part of the paper.
>
> > **Line 179: I'd suggest using...**
>
> In choosing the notations, we were guided by the traditions of previous works. It is tacitly accepted that the accuracy of the problem solution is denoted as $\varepsilon$ or $\epsilon$. Since in our work we are essentially solving two optimization problems, we have chosen these two notations accordingly. However, if you recommend the most appropriate notation for solving the inner optimization problem (linear search), we will be happy to change the notation.
>
> > **In (8), it would be better...**
>
> Thank you for this comment, we will correct this typo.
>
> > **Why is coordinate descent the best thing to do? To me...**
>
> First of all, we would like to mention that in previous works (with the Order Oracle) the authors used various schemes (including the transition to the normalized gradient algorithm), but all these works, including a very recent paper [1] (which appeared after our submission to the conference) achieved only unaccelerated estimates, moreover, all these estimates contain explicit dimensionality. This fact demonstrates that it has not yet been possible to find such an accelerated method that is satisfied with only the direction of the gradient (without knowledge of the modulus)... This includes among various methods with auxiliary low-dimensional minimization [2-4]. In all the accelerated methods known to us, the computation of the real gradient was assumed in one way or another. Therefore, the approach proposed in our paper is perhaps the only feasible scenario at the moment. Moreover, prior to our work, it was not clear that it was possible in any existing accelerated coordinate algorithm **to completely get rid of the knowledge of directional derivatives** and reduce everything to one-dimensional searches. It is worth noting that this observation is not trivial. For example, this was not the case with full-gradient accelerated algorithms. Moreover, we did not immediately succeed with coordinate algorithms, since there are already many of them. We had to find one for which it was possible to prove the corresponding statement.
>
> Regarding non-accelerated methods, our approach explicitly demonstrates the following: we achieve SOTA results for a class of unaccelerated algorithms up to logarithm factor (see Table 1). Moreover, since the dimensionality dependence is hidden in parameters such as $S_{\alpha}$, in some cases (e.g., *the asymmetric case*: when the trace of the Hessian matrix has the same order as the largest eigenvalue) this dependence is reduced (this is typical for the class of coordinate descent), thus outperforming previous work in the high-dimensional problem.
>
> Moreover, we did not stop there and provided an algorithm that improves the already accelerated convergence estimates in the case of the low-dimensional problem (see Appendix G). Finally, in Section 5 of our paper, we opened a new research direction (stochastic order oracle) by providing the first convergence results of another algorithm, but already in the stochastic oracle concept.
>
> > **Algorithm 1 as stated works...**
>
> As we mention in Section 3, in the problem formulation, the stochasticity $\xi$ for Algorithm 1 and 2 denotes the $i$-th coordinate, and $\mathcal{D}$ represents the distribution $p_{\alpha}(i)$. But for the algorithm described in Section 5, stochasticity means the $\xi$-th realization of the function.
>
> > **The accelerated algorithm 2 has several loops...**
>
> Regarding the use of the second linear search, it is currently necessary to use the linear search twice per iteration for the theoretical proof of estimates. But the good news is that in practical experiments (all the ones we have encountered), using only the first linear search allows us not only to significantly reduce the running time of the algorithm, but also to improve the convergence itself (see Figure 4). Regarding the noise $\delta(x,y)$, in this paper, for simplicity of presentation, we omitted it (we assumed that it does not exist), since our goal was to show that acceleration in such an oracle concept exists. However, the analysis for the case when there is noise will not be very different from the proposed one. We left this technical moment as a future work.
>
> > **Similarly in Section 5, these reductions to existing methods...**
>
> In order to talk about the optimality of certain algorithms, we need to compare our upper estimates with the lower ones. However, as far as we know, there are currently no lower estimates for algorithms using the Order Oracle. So now we can only compare with existing SOTA results and assume that these estimates are likely to be optimal.
>
> [1] Zhang, C., & Li, T. (2024). Comparisons Are All You Need for Optimizing Smooth Functions. arXiv preprint arXiv:2405.11454.
>
> [2] Drori, Y., & Taylor, A. B. (2020). Efficient first-order methods for convex minimization: a constructive approach. Mathematical Programming, 184(1), 183-220.
>
> [3] Narkiss, G., & Zibulevsky, M. (2005). SESOP: Sequential Subspace Optimization Method.
>
> [4] Hager, W. W., & Zhang, H. (2006). A survey of nonlinear conjugate gradient methods. Pacific journal of Optimization, 2(1), 35-58.
>
>
> With Respect,
>
> Authors

---

> > ### Comment · Reviewer_ErqJ · 2024-08-10
> >
> > Thanks for the response and for explaining.
> >
> > I remain somewhat neural to the paper / contribution for the following reason: it is straightforward that given any random (i.e., importance-sampling type) coordinate algorithm, such as (3), a line search greedy step can only guarantee more progress, and it is because of the fact the algorithm is built on coordinate update that the 1D binary search is efficient. The same trick, of course, can be applied to coordinate-based accelerated method, which would now require knowledge of $L_i,\mu...$ to implement.
> >
> > There seems to be various hidden $d$ dependence in the $S$'s and $L_i$'s: There was a remark about "when the trace of the Hessian matrix has the same order as the largest eigenvalue, this dependence on $d$ is reduced" in which case the problem itself sounds like a 1D-ish problem to me. The methodology feels a bit like a tag-on to existing methods - I think the paper would benefit from a more thorough study of when/why this is the best "gradient estimator" vs. alternatives potentially using full-gradient-like information.

---

> ### Author Response · Authors · 2024-08-10
>
> Dear **Reviewer ErqJ**,
>
> We thank you for your prompt feedback! Perhaps we have not clearly described the full importance of our results in Rebuttal, so with all due respect to you, **we would like to reiterate the full novelty and significance of our results** for the optimization community and beyond.
>
> We would like to point out that our work focuses on the case where the oracle cannot return the objective function values $f(x)$ at the requested point $x$, much less the function gradient. Our oracle can only compare the objective function values (see the definition of Order Oracle (2)). There are a number of works in such an oracle concept (see Table 1). As shown in Table 1, the research on the optimization problem with such an oracle is at least c 2019 and **the question of the existence of an accelerated algorithm in such oracle concept remained open until our work**. For example, in [1] an attempt was made to create an accelerated algorithm using the heavy ball momentum, but the result was still an unaccelerated algorithm. Moreover, as mentioned earlier above, a recent paper [2] (which appeared after our conference submission) proposes another (alternative) approach that also does not produce an accelerated algorithm in the Order Oracle concept.
>
> Thus, one of the main contributions of our paper is the proposed new approach to create (both unaccelerated and **accelerated**) optimization algorithms (1) using only Order Oracle (2). We agree that our approach for creating unaccelerated algorithms is quite simple, but we have demonstrated the effectiveness of this approach by proposing an alternative to the existing ones. But the value of the approach proposed in our work demonstrates the possibility of creating an accelerated algorithm (using Order Oracle).
>
> >**The same trick, of course...**
>
> We cannot agree with this statement. Unlike the unaccelerated algorithm, **it is not obvious** that a linear search can be applied to the accelerated method of coordinate descent (see any accelerated algorithm of coordinate descent). Moreover, it is even more not obvious, and even surprising, that to create an accelerated algorithm with Order Oracle, it is necessary to use not one linear search, but two. As we said, there are a number of accelerated coordinate descent algorithms, and one of the problems we faced while preparing this article was to find the most appropriate accelerated coordinate descent algorithm on the basis of which we could create an already accelerated algorithm with Order Oracle.
>
> >**There seems to be various hidden $d$ dependence...**
>
> As we have already mentioned, we have retained the hidden dimensionality dependence from the coordinate algorithm class. Indeed, comparable to other algorithms using Order Oracle concepts (see Related Work, namely "Algorithms with Order Oracle"), there are cases where hidden dimensionality dependence demonstrates superiority over explicit dependence.
>
> In our opinion, the proposed approach demonstrating the first accelerated algorithm in the Order Oracle concept **is no longer a minor contribution to the paper. Nevertheless, we also have equally significant results**, namely, we show how to improve the estimates of the proposed accelerated algorithm with Order Oracle (the low-dimensional problem case). Finally, we provide the first convergence result of the algorithm using the already stochastic concept of Order Oracle (see the definition in (8)).
>
> [1] Gorbunov, E. et. al (2019, September). A Stochastic Derivative Free Optimization Method with Momentum. In International Conference on Learning Representations.
>
> [2] Zhang, C., & Li, T. (2024). Comparisons Are All You Need for Optimizing Smooth Functions. arXiv preprint arXiv:2405.11454.
>
>
> With Respect,
>
> Authors

---

### Official Review · Reviewer_WKzP · 2024-07-08

**Soundness:** 3
**Presentation:** 3
**Contribution:** 3
**Rating:** 5
**Confidence:** 3

**Summary:**

This paper addresses challenges in black-box optimization by introducing the concept of an Order Oracle. This oracle compares function values without assuming access to their actual values, allowing for the creation of new optimization algorithms. The paper proposes both deterministic and stochastic approaches, achieving SOTA convergence results and demonstrating the effectiveness of these algorithms through numerical experiments.

**Strengths:**

1. The order oracle setting is new.
2. The theoretical analysis seems solid.

**Weaknesses:**

1. Elaboration on Motivation: While the setting of the problem is interesting, the motivation for addressing this particular problem could be further elaborated, especially in the introduction. Providing a more detailed context and explanation of the significance and potential impact of solving this problem would help readers better understand its importance.

2. Comprehensive Experimental Validation: The experimental validation appears less comprehensive, as it relies on only one simple function. It would be much more convincing to apply the proposed method to the motivating problem outlined in the introduction. This would demonstrate the method's practical applicability and robustness in a real-world scenario.

3. Performance in High-Dimensional Space: I am curious about the performance of the proposed method in high-dimensional space, as this is often a significant challenge for coordinate descent methods. Could the author provide more insights or experimental results that address this aspect? If there are inherent limitations related to high-dimensional settings, it would be beneficial to discuss and highlight these issues. Providing such information would offer a clearer understanding of the method's strengths and potential limitations.

**Questions:**

1. Technical Challenges compared to Nesterov's 2012 Work: I am curious about the specific technical challenges involved in the proof, especially in relation to extending and building upon Nesterov's 2012 work. Could the author provide more detail on the difficulties encountered and the novel techniques or modifications that were necessary to address these challenges?

2. Dimension-Independent Complexity: I notice that the paper presents a dimension-independent ($d$) complexity in comparison with existing works, as shown in Table 1. This is a significant and surprising result. Could the author elaborate more on this aspect? Specifically, I am interested in understanding the implications of achieving dimension-independent complexity. Or the dependence is manifested in some parameters.

**Limitations:**

The author could discuss the limitations in high-dimensional space if this is indeed a constraint of the proposed method.

---

> ### Author Rebuttal · Authors · 2024-08-04
>
> Dear **Reviewer WKzP**,
>
> We thank you for your feedback on our work. We provide detailed answers to the comments and questions raised in the review below.
>
> > **Elaboration on Motivation.**
>
> Despite the potential motivation given in the introduction, as well as the mention of Appendix A, which details an application (startup: smart coffee machine), we are convinced that problem setting with such an oracle has a huge number of potential applications, in particular, such an oracle is organic to perhaps the most popular setting at the moment, namely *Reinforcement learning with Human Feedback (RLHF)*. And we agree that the paper will become stronger and potentially have even more impact when we add more mentions of RLHF in the motivational part of the main paper.
>
> > **Comprehensive Experimental Validation.**
>
> Indeed, this paper was initiated due to a challenge faced by one of the co-authors in the realization of a startup (a smart coffee machine). Already in the process of preparing the paper, we started testing it. At the moment we have ready experiments on more than a real problem, where the coffee machine uses the algorithms presented in this paper, with a deterministic oracle. However, we have encountered some difficulties: since the coffee machine is not yet patented (we've started the patenting process), we have not been able to insert these experiments. Regardless, we thought we found a decent temporary alternative - **model examples that clearly verify the theoretical results**. Nevertheless, we realize that adding these results will further strengthen our work. Currently we expect the patenting process for the coffee machine to be completed in September-October this year, so in case of a positive decision (acceptance at the conference) on our paper, we are confident that we will have time to add the experiments on real problems to the camera-ready version.
>
> > **Performance in High-Dimensional Space.**
>
> The high-dimensional challenge problem is typical for a class of coordinate algorithms. And since we are based on such an algorithm, we expectedly inherit this challenge. However, as you have already noticed below (in the Questions section), there is no explicit dependence on dimensionality in the final estimates (unlike previous work with Order Oracle). This fact is an advantage of our scheme (approach), since there are cases (e.g., *the asymmetric case*: when the trace of the Hessian matrix has the same order as the largest eigenvalue) where the dimensionality dependence is reduced (again, this is typical for the class of coordinate algorithms). In this case, our algorithm will be significantly superior to all previous algorithms, since in alternative algorithms the explicit dependence on dimension $d$ does not disappear anywhere.
>
> > **Technical Challenges compared to Nesterov's 2012 Work.**
>
> To begin with, we would like to clarify that indeed the optimization problem formulation is somewhat similar to the one considered in [1], with perhaps a serious exception: we cannot use information about the function (Order Oracle concept).  As can be seen from Table 1, all previous optimization algorithms using the Order Oracle achieved only unaccelerated convergence estimates. Moreover, a recent paper [2], which appeared on the arXiv after our submission to the conference also provided only an unaccelerated algorithm. This fact demonstrates that it has not yet been possible to find such an accelerated method that is satisfied with only the direction of the gradient (without knowledge of the modulus)... This includes among various methods with auxiliary low-dimensional minimization [3-5]. In all the accelerated methods known to us, the computation of the real gradient was assumed in one way or another. *Therefore, it is important to take into account the following*:
> - The approach proposed in our paper is perhaps the only feasible scenario at the moment.
> - Prior to our work, it was not clear that it was possible in any existing accelerated coordinate algorithm **to completely get rid of the knowledge of directional derivatives** and reduce everything to one-dimensional searches. It is worth noting that this observation is not trivial. For example, this was not the case with full-gradient accelerated algorithms.
> - We did not immediately succeed with coordinate algorithms, since there are already many of them. We had to find one for which it was possible to prove the corresponding statement.
>
> However, we did not stop there and provide an algorithm (see Appendix G) that shows the possibility of improving convergence estimates in the case of a low-dimensional optimization problem. Finally, we opened a new research direction by providing the first convergence results for an optimization algorithm with a stochastic Order Oracle (see Section 5), which uses a different approach.
>
> > **Dimension-Independent Complexity.**
>
> This is an interesting question! Yes, by using unusual sampling of the active coordinate, the dependence on dimensionality is implicitly hidden in parameters such as $S_{\alpha} = \sum_{i}^{d} L_{i}^{\alpha}$. For example, if we consider a uniform distribution ($\alpha = 0$), then $S_0 = d$. Also, we propagate the best results among accelerated coordinate descent algorithms by using the following distribution to select the active coordinate: $i=\mathcal{R_{\alpha/2}}(L)$. This is to get rid of the dimension dependence in the estimates, since $S_{\alpha/2} \leq \sqrt{d S_{\alpha}}$.
>
> [1] Nesterov, Y. (2012). Efficiency of coordinate descent methods on huge-scale optimization problems.
>
> [2] Zhang, C. et al. (2024). Comparisons Are All You Need for Optimizing Smooth Functions.
>
> [3] Drori, Y. et al. (2020). Efficient first-order methods for convex minimization: a constructive approach.
>
> [4] Narkiss, G. et al. (2005). SESOP: Sequential Subspace Optimization Method.
>
> [5] Hager, W. et al. (2006). A survey of nonlinear conjugate gradient methods.
>
>
> With Respect,
>
> Authors

---

> > ### Comment · Reviewer_WKzP · 2024-08-11
> >
> > I appreciate the authors' thorough responses to my questions. After carefully reviewing their answers, I have no further questions at this stage and will provide my feedback to the AC in the next phase.

---

> > > ### Author Response · Authors · 2024-08-12
> > >
> > > Dear **Reviewer WKzP**,
> > >
> > > We thank you for taking the time to check our responses.
> > >
> > > With Respect,
> > >
> > > Authors

---

### Official Review · Reviewer_uSy5 · 2024-07-10

**Soundness:** 3
**Presentation:** 3
**Contribution:** 2
**Rating:** 5
**Confidence:** 3

**Summary:**

this paper focuses on solving the black-box optimization problem with order oracle method. specifically, the author provided a non-accelerated deterministic optimization algorithm, which relies on weighted sampling of the coordinate and the GRM method to resolve a linear search subproblem. the author provided convergence analysis on non-convex, convex and strongly convex cases and showing a superior iteration complexity. the author further an accelerated method called OrderACDM and proved its linear convergence. the stochastic case is discussed with asymptotic convergence results. Finally, the author provided a very simple numerical experiments to show the advantages of the proposed algorithms.

**Strengths:**

1. overall, the paper is clear and easy to follow.
2. the author provided solid analysis to the convergence of the proposed algorithms and the results look correct to me.
3. the proposed algorithm looks clean and simple.

**Weaknesses:**

1. i have some concerns on the novelty of this paper. in the non-accelerated algorithm, the two main techniques (the weighted sampling and the GRM) are not new. and the accelerated algorithm closely follows ( Nesterov and Stich, 2017).
2. both of the proposed algorithms are double loop algorithms as in each iteration, a linear search sub-problem is solved by GRM. as the author mentioned, GRM requires O(log 1/eps) iterations, which should be considered significant especially in the case of strongly convex problem. the author is hiding this factor with \tilde{O} in table 1, which makes the comparison unfair.
3. the numerical experiment is too simple. i understand this is a theoretical paper, but the experiments on a toy example is still below the bar of acception.
4. in the accelerated case, it's not clear to me how the proposed algorithm is compared with  (ACDM), Nesterov and Stich, 2017) and (NU-ACDM), Allen-Zhu et al., 2016). the proposed algorithm looks slower to me since it is double loop. i would suggest the author to add more discussions on the convergence results.


minor issues:

1. line 77, missing something after "to the inner product as.." ?
2. line 78, L_i appears before it is defined.
3. table 1, many parameters appear before they are defined.

**Questions:**

1. in assumption 1.1, can L-coordinate-Lipschitz suggest L-smooth?

**Limitations:**

see above

---

> ### Author Rebuttal · Authors · 2024-08-04
>
> Dear **Reviewer uSy5**,
>
> We thank you for your interest in our work. We attach below detailed responses to the comments and questions raised in the review.
>
> > **i have some concerns on the novelty of this paper...**
>
> We tried to prepare the text of the paper so that it would be easily accessible to readers not only for the optimization community, as we believe that the results of the paper can have an impact on various areas. In particular, perhaps the most popular area at the moment is *Reinforcement learning with Human Feedback (RLHF)*.
>
> Perhaps it is precisely because of the simplicity and accessibility of the presentation of the paper that the misunderstanding arose. With all due respect to you, **we would like to emphasize the significance and novelty of our work**. As can be seen from Table 1, all previous optimization algorithms using the Order Oracle achieved only unaccelerated convergence estimates. Moreover, a recent paper [1], which appeared on the arXiv after our submission to the conference also provided only an unaccelerated algorithm. This fact demonstrates that it has not yet been possible to find such an accelerated method that is satisfied with only the direction of the gradient (without knowledge of the modulus)... This includes among various methods with auxiliary low-dimensional minimization [2-4]. In all the accelerated methods known to us, the computation of the real gradient was assumed in one way or another. Therefore, it is important to take into account the following:
> - The approach proposed in our paper is perhaps the only feasible scenario at the moment.
> - Prior to our work, it was not clear that it was possible in any existing accelerated coordinate algorithm **to completely get rid of the knowledge of directional derivatives** and reduce everything to linear search. It is worth noting that this observation is not trivial. For example, this was not the case with full-gradient accelerated algorithms.
> - We did not immediately succeed with coordinate algorithms, since there are already many of them. We had to find one for which it was possible to prove the corresponding statement.
>
> However, we did`t stop there and provide an algorithm (see Appendix G) that shows the possibility of improving convergence estimates in the case of a low-dimensional optimization problem. Finally, we opened a new research direction by providing the first convergence results for an optimization algorithm with a stochastic Order Oracle (see Section 5), which uses a different approach. As can be seen, all of these results individually, and even more so in combination, are novel and are especially needed in various applications.
>
> > **both of the proposed algorithms are...**
>
> We do not quite agree with this remark, since for us the notation $\tilde{O}(\cdot)$ is equivalent to explicitly writing the logarithm in the estimates. Moreover, in Section 1.2 (Notation) we introduced this notation ($\tilde{O}(\cdot)$, where we explained that it hides the logarithmic coefficient, which is standard among optimization works) to reduce the length of the formulas, in particular in Table 1. To be fair, we spell out the presence of the logarithm in the estimates in all discussions of the results, and even more so in the abstract of the paper. However, if you strongly recommend explicitly stating the logarithm in the estimates, we will of course do so.
>
> > **the numerical experiment is too simple...**
>
> As written in Appendix A1, this paper was initiated due to a challenge faced by one of the co-authors in the realization of a startup (a smart coffee machine). Already in the process of preparing the paper, we started testing it. At the moment we have ready experiments on more than a real problem, where the coffee machine uses the algorithms presented in this paper, with a deterministic oracle. However, we have encountered some difficulties: since the coffee machine is not yet patented (we've started the patenting process), we have not been able to insert these experiments. Regardless, we thought we found a decent temporary alternative - **model examples that clearly verify the theoretical results**. Nevertheless, we realize that adding these results will further strengthen our work. Currently we expect the patenting process for the coffee machine to be completed in September-October this year, so in case of a positive decision (acceptance at the conference) on our paper, we are confident that we will have time to add the experiments on real problems to the camera-ready version.
>
> > **in the accelerated case, it's not clear to me...**
>
> We would like to clarify that all figures show convergence depending on the iteration complexity. As shown by theoretical evaluations, the iteration complexities of the proposed algorithms match those of the coordinate descent (first order) algorithms, but already the oracle complexities lose by a logarithmic factor. We mention this in the discussion under each result. Moreover, in Figure 4, we compare the proposed accelerated algorithm with the ACDM algorithm of [5]. In Figure 4, we show that in numerical experiments, using only the first linear search not only improves the running speed of the algorithm, but also improves the convergence itself.
>
> > **minor issues.**
>
> Thank you, we will correct those typos.
>
> > **in assumption 1.1, can...**
>
> Assumption 1.1 is standard for the class of coordintate algorithms and implies some L-smooth over each coordinates.
>
> [1] Zhang, C. et al. (2024). Comparisons Are All You Need for Optimizing Smooth Functions.
>
> [2] Drori, Y. et al. (2020). Efficient first-order methods for convex minimization: a constructive approach.
>
> [3] Narkiss, G. et al. (2005). SESOP: Sequential Subspace Optimization Method.
>
> [4] Hager, W. W. et al. (2006). A survey of nonlinear conjugate gradient methods.
>
> [5] Nesterov, Y. et al. (2017). Efficiency of the accelerated coordinate descent method on structured optimization problems.
>
> With Respect,
>
> Authors

---

> > ### Comment · Reviewer_uSy5 · 2024-08-12
> > **reply to authors**
> >
> > I thank the authors for addressing my questions and concerns. Now i agree that the main contribution in this paper is to deal with a special case where only the order of obj function values can be accessed. I have increased my score to be 5. however, my main concern is still the design of the double loop algorithm that leads to an additional log factor and the experiments part. the author mentioned that they could add more practical experiment results. but those results may need another round of review in order to be considered.

---

> > > ### Author Response · Authors · 2024-08-13
> > >
> > > Dear **Reviewer uSy5**,
> > >
> > > We are grateful to the Reviewer for checking our responses and for raising the score! However, since there are still concerns about our work, we would like to provide detailed answers:
> > >
> > > >**however, my main concern is still the design of the double loop algorithm that leads to an additional log factor**
> > >
> > > The number of oracle calls per iteration possibly can be further reduced in log factor... However, **firstly**, such a method (for our oracle) is simply not known. And **secondly**, the total number of oracle calls (Order Oracle calls) may even be better with our approach, because the iterations number may become smaller due to one-dimensional searches. In any case, this is observed for accelerated methods with low-dimensional auxiliary searches (*by the way, note that our oracle could not be integrated into such methods!*): [1] (see experiments and references to other works at the end of the paper), and [2]. Nevertheless, we have shown in Figure 6 (see Appendix B) that using only the first linear search not only reduces computational resources but also improves convergence. Based on our observation, we may well expect as future results an improvement of our estimate by at least one logarithm.
> > >
> > > >**and the experiments part. the author mentioned that they could add more practical experiment results. but those results may need another round of review in order to be considered.**
> > >
> > > We would like to emphasize that our work is mainly theoretical. Our main results are four new algorithms: an unaccelerated and accelerated algorithm that uses a deterministic Order Oracle (see Section 3 and 4, respectively), the asymptotic convergence of an algorithm that already uses a stochastic Order Oracle (see Section 5), and finally an algorithm that shows how the convergence results of the accelerated algorithm can be improved given a low-dimensional problem (see Appendix G). All these results **individually open directions for future research**, showing an already complete and, more importantly, meaningful paper. However, we realize that our problem statement has many potential applications, so, as we said, we will definitely add experiments on a real-world problem (the smart coffee machine - a startup) but in Appendix (**since our results are fundamental and not limited to a startup**). Nevertheless, *we believe that our current experimental part is fully in the spirit of the theoretical work*.
> > >
> > > Finally, we would like to thank the **Reviewer uSy5** once again for taking the time to check our responses. We hope that with this answer we have solved all the remaining concerns of the Reviewer. *However, if you have any further questions, we will be happy to answer them!*
> > >
> > > [1] Guminov, S., Gasnikov, A., & Kuruzov, I. (2023). Accelerated methods for weakly-quasi-convex optimization problems. Computational Management Science, 20(1), 36.
> > >
> > > [2] Drori, Y., & Taylor, A. B. (2020). Efficient first-order methods for convex minimization: a constructive approach. Mathematical Programming, 184(1), 183-220.
> > >
> > > Best regards,
> > >
> > > Authors

---

### Official Review · Reviewer_K16N · 2024-07-11

**Soundness:** 4
**Presentation:** 3
**Contribution:** 3
**Rating:** 7
**Confidence:** 4

**Summary:**

The paper considers the “Order Oracle” for optimization, which does not require the function values or gradient information, but only the relative ordering of the function values at different points. This can also be done with some small noise. This model captures the challenges encountered in real-world black box optimization problems and the authors motivate it via the ongoing developments in generative models. The main contributions of the paper are:

 1. The paper provides SOTA convergence results up to log factors for deterministic algorithms with order oracle in the non-convex, convex and strongly convex settings.
2. The paper shows that acceleration in such an oracle is possible by giving an algorithm for strongly convex functions. They further show that the convergence results can be improved when the problem is low dimensional and compare it to the ellipsoid method.
3. The paper gives a stochastic version of the algorithm.

The accelerated and unaccelerated algorithms are inspired by coordinate descent methods. The method is essentially coordinate descent where the step size is chosen via a specific line search procedure (golden ratio method) which is implemented using the order oracle.

For the accelerated algorithm, the authors need to include a secondary line search to ensure that the iterates are always making progress.

**Strengths:**

The paper demonstrates that using a very weak oracle such as order oracle is sufficient to get rates of convergence that match the rates of the optimal gradient methods via first order oracles. The algorithms are quite simple and work well in practice as shown in the experiments. The ideas are original and the results are presented clearly.

**Weaknesses:**

The authors have motivated the use of this oracle, but it would be useful to have some concrete examples on how is it being used in practice.

**Questions:**

1. When you say adaptive in the text, what does that mean?
2. The algorithms basically seem like you are performing coordinate descent, but instead of using gradients* step size, you want to directly compute this quantity using the GRM line search. So implicitly this is a first order method being implemented using an order oracle? If this is the case, is it possible to show an equivalence between the order oracle and gradient oracle?
3. What is the notation with $\alpha$?
4. Are there examples (even experiments) that can say something about the difference between the order oracle and usual first order oracles?
5. In the accelerated algorithm, are there adversarial examples where the second line search is necessary?
6. What is the barrier in extending the acceleration beyond strongly convex objectives.

**Limitations:**

The paper considers a weaker oracle model than what is usually considered in standard optimization results and gives algorithms under this oracle. It is not clear what the limitations of the oracle are from the paper, but the results match the ones obtained via stronger gradient oracles.

---

> ### Author Rebuttal · Authors · 2024-08-04
>
> Dear **Reviewer K16N**,
>
> We thank you for your positive evaluation of our work! Below we provide detailed answers to all comments and questions that arose in the review.
>
> > **The authors have motivated the use of this oracle, but it would be useful to have some concrete examples on how is it being used in practice.**
>
> With all due respect to you, we do not fully agree with this comment, as a more concrete example given in Appendix A1 (startup: smart coffee machine) seems hard to think of. But despite the fact that this research was indeed initiated due to a problem faced by one of the co-authors in implementing a startup, we are convinced that problem setting with such an oracle has a huge number of potential applications, in particular, such an oracle is organic in perhaps the most popular setting at the moment, namely Reinforcement learning with Human Feedback (RLHF). And we agree that the article will become stronger and potentially have even more impact when we add more mentions of RLHF in the motivational part of the main paper.
>
> > **When you say adaptive in the text, what does that mean?**
>
> This is a really interesting question! The unaccelerated algorithm presented in Section 3 can be classified as an adaptive algorithm (especially in the case of uniformly distributed selection of the active coordinate $\alpha = 0$), since the iteration step and the value of the gradient coordinate are found without any parameters (in particular without the $L$-coordinate-Lipschitz constants) by solving an internal linear problem. This adaptability is clearly shown in Figure 2, where OrderRCD outperforms the first-order RCD algorithm, which uses the gradient coordinate with a theoretical step value using the constant $L_i$. That is, the advantage of our algorithm arises because we adaptively select the iteration step of the algorithm in contrast to the first-order algorithm.
>
> > **The algorithms basically seem like you are performing coordinate descent, but instead of using gradients step size, you want to directly compute this quantity using the GRM line search. So implicitly this is a first order method being implemented using an order oracle? If this is the case, is it possible to show an equivalence between the order oracle and gradient oracle?**
>
> Indeed, the class of coordinate algorithms was chosen as a basis because the issue of not knowing the function gradient was organically solved by using the golden ratio method (linear search), where the Order Oracle is already used. Therefore, our algorithm almost completely adopted both all the advantages of the coordinate algorithm (which was our goal) and the disadvantages. Moreover, as often mentioned in this paper, we cannot say that these algorithms are equivalent, because the oracle complexity differs by a logarithm. This is the cost of using linear search.
>
> > **What is the notation with $\alpha$?**
>
> By $\alpha$ we mean that this is the parameter used in generating the active coordinate $i_k$ with the following distribution: $p_{\alpha}(i) = L_i^{\alpha} / S_\alpha,$ where $i \in [d]$, and $S_{\alpha} = \sum_{i}^{d} L_i^{\alpha}$. This choice of active coordinate is already standard in works devoted to coordinate descent and is actively used in various works, e.g., [1-4].
>
> > **Are there examples (even experiments) that can say something about the difference between the order oracle and usual first order oracles?**
>
> Yes, Figure 2 and 4 show the comparison of unaccelerated and accelerated algorithms respectively. The algorithms proposed in this paper (Order Oracle) are compared with SOTA coordinate descent algorithms (first order oracle). It is worth noting, as discussed above, the unaccelerated algorithm with Order Oracle can quite expectedly outperform the first order algorithm (RCD) due to adaptivity. But the accelerated algorithm with Order Oracle with the given recommendations concerning the number of linear searches is not inferior to the first-order algorithm.
>
> > **In the accelerated algorithm, are there adversarial examples where the second line search is necessary?**
>
> At the moment it is necessary to use linear search twice per iteration for theoretical proof of estimates. This is basically not bad considering that our goal was to show the possibility of acceleration. But the good news is that in practical experiments (all the ones we have encountered), using only the first linear search allows us not only to significantly reduce the running time of the algorithm, but also to improve the convergence itself (see Figure 4).
>
> > **What is the barrier in extending the acceleration beyond strongly convex objectives?**
>
> As already mentioned, one of the main goals of our work is to show the existence/possibility of accelerating algorithms using the concept of the Order Oracle. We have demonstrated this on the example of a strongly convex function. By doing so, we have opened a direction for future work, in particular, to investigate already accelerated algorithms under other assumptions. Regarding obtaining convergence results of the accelerated algorithm in the convex smooth case, this is possible using standard tricks such as, for example, regularization. We left further development of the results as future work.
>
> [1] Bubeck, S. (2015). Convex optimization: Algorithms and complexity. Foundations and Trends® in Machine Learning.
>
> [2] Nesterov, Y. (2012). Efficiency of coordinate descent methods on huge-scale optimization problems. SIAM Journal on Optimization.
>
> [3] Nesterov, Y., & Stich, S. U. (2017). Efficiency of the accelerated coordinate descent method on structured optimization problems. SIAM Journal on Optimization.
>
> [4] Allen-Zhu, Z., Qu, Z., Richtárik, P., & Yuan, Y. (2016, June). Even faster accelerated coordinate descent using non-uniform sampling. In International Conference on Machine Learning. PMLR.
>
> With Respect,
>
> Authors

---

> > ### Comment · Reviewer_K16N · 2024-08-11
> >
> > Thanks for the clarifications and answers to my questions! I hope the ones that are missing from the paper end up in the next version!

---

> > > ### Author Response · Authors · 2024-08-12
> > >
> > > Dear **Reviewer K16N**,
> > >
> > > We thank you for checking our response and for your very positive evaluation.
> > >
> > > With Respect,
> > >
> > > Authors

---

### Official Review · Reviewer_XqbR · 2024-07-12

**Soundness:** 3
**Presentation:** 2
**Contribution:** 3
**Rating:** 6
**Confidence:** 4

**Summary:**

This paper explores the use of a zero-order oracle called the Order Oracle to solve optimization problems where exact objective function values are unknown. This oracle focuses on comparing the order of objective function values rather than requiring exact values.  The authors propose new non-accelerated algorithm, OrderRCD, that integrate the Order Oracle into coordinate descent algorithm and achieves performance comparable to state-of-the-art methods in non-convex, convex, and strongly convex settings. This involves randomly selecting coordinates and performing linear searches to find optimal step sizes. They also  present an accelerated optimization algorithm called OrderACDM, which uses two linear searches in each iteration to adaptively determine optimal step sizes. They show faster convergence rates and improved efficiency in strongly convex setting. The paper extends the concept to stochastic settings, showing asymptotic convergence. Numerical experiments validate the theoretical results, demonstrating that OrderRCD performs comparably to traditional methods, while OrderACDM converges faster, outperforming other methods.

**Strengths:**

The paper formalizes a novel concept called Order Oracle, which enables optimization without requiring exact function values.  Building on this concept, the authors develop two new algorithms, OrderRCD and OrderACDM, which integrate the Order Oracle into the coordinate descent framework. The paper provides a  comprehensive theoretical analysis demonstrating the effectiveness of OrderRCD across non-convex, convex, and strongly convex settings and showing faster convergence rate of OrderACDM in strongly convex setting.  The convergence proofs of both the algorithms are detailed. Additionally, this paper also introduce the first algorithm using the stochastic Order Oracle with asymptotic convergence guarantees. They also show practical effectiveness of OrderRCD and OrderACDM compared to traditional methods through their numerical experiments.

**Weaknesses:**

The numerical experiments show the evaluation of OrderRCD and OrderACDM in standard quadratic functions. Experiments on real-world problems would strengthen the validation. The paper can include specific details about the numerical experiments, the exact values or structure of matrices and vectors $A$, $b$, and $c$, the dimensionality of the problems tested, datasets used etc.  The paper does not deeply explore the scalability of the proposed algorithms in very high-dimensional settings or large-scale datasets.  While the paper addresses stochastic optimization, analysis of noise sensitivity would enhance the understanding of the algorithm's performance under various conditions.

**Questions:**

1. Could you provide more specifics about the numerical experiments?
2.  Could you show numerical validation in real-world problems and datasets?
3. Are there any experiments being conducted for the stochastic order oracle? Could you show the results in that setting?

**Limitations:**

1. The theoretical guarantees  rely heavily on the assumptions of strong convexity of the objective function. These assumptions may not hold for all types of optimization problems, limiting the applicability of the proposed methods to broader, less structured scenarios.
2. The paper does not provide practical demonstrations on large-scale, high-dimensional datasets. Additional experiments on a diverse set of objective functions and real-world datasets would provide a more comprehensive validation of the algorithms' performance and robustness
3. The accelerated version requires updating multiple parameters and has multiple steps and iterative processes that potentially increase the computational load.
4. In large-scale problems with high-dimensional data, the computational overhead from linear searches, parameter updates, and coordinate selections can become significant. The scalability of the algorithm is thus a concern, and its performance may degrade as the problem size increases.

---

> ### Author Rebuttal · Authors · 2024-08-04
>
> Dear **Reviewer XqbR**,
>
> We would like to thank you for your time for preparing the review.
>
> > **The numerical experiments show the evaluation of OrderRCD and OrderACDM in standard quadratic functions. Experiments on real-world problems would strengthen the validation. The paper can include specific details about the numerical experiments, the exact values or structure of matrices and vectors $A$, $b$, and $c$, the dimensionality of the problems tested, datasets used etc. The paper does not deeply explore the scalability of the proposed algorithms in very high-dimensional settings or large-scale datasets. While the paper addresses stochastic optimization, analysis of noise sensitivity would enhance the understanding of the algorithm's performance under various conditions.**
>
> Since the presented questions fully characterize the weaknesses mentioned in the review, we have prepared detailed answers to each of the questions. **The answers can be found below.**
>
> > **Could you provide more specifics about the numerical experiments?**
>
> First, we would like to clarify that due to the lack of space in the main part of the paper, several numerical experiments can be found also in Appendix B. Regarding the reproducibility of the results, in our opinion it seems that there is enough information about the problem formulation and the parameters of the algorithm. However, we agree that we could be a bit more precise in describing the matrix $A$ and vectors $b,c$, namely the process of their generation. We will add this information about the problem formulation to the main part of the paper.
>
> > **Could you show numerical validation in real-world problems and datasets?**
>
> As written in Appendix A1, this paper was initiated due to a challenge faced by one of the co-authors in the realization of a startup (a smart coffee machine). Already in the process of preparing the paper, we started testing it. At the moment we have ready experiments on more than a real problem, where the coffee machine uses the algorithms presented in this paper, with a deterministic oracle. However, we have encountered some difficulties: since the coffee machine is not yet patented (we've started the patenting process), we have not been able to insert these experiments. Regardless, we thought we found a decent temporary alternative - **model examples that clearly verify the theoretical results**. Nevertheless, we realize that adding these results will further strengthen our work. Currently we expect the patenting process for the coffee machine to be completed in September-October this year, so in case of a positive decision (acceptance at the conference) on our paper, we are confident that we will have time to add the experiments on real problems to the camera-ready version.
>
> >**Are there any experiments being conducted for the stochastic order oracle? Could you show the results in that setting?**
>
> Of course, yes, they are being conducted! To be more specific, there are tests of the coffee machine at the moment, where the ideal coffee is created already for a certain group of people on average. We also want to add the results of the experiments to the Appendix section in the final version of the paper after obtaining a patent for the coffee machine.
>
> >**Limitations.**
>
> Thank you, we will be sure to add a "Limitations" section to the main part of the paper.
>
> # **Significance and novelty of our results**
> In conclusion, as already mentioned, this work was motivated by a specific application (startup), but the results presented are not limited to it! This oracle concept has a number of applications, including perhaps the most popular setup, namely Reinforcement learning with Human Feedback (RLHF). It is worth noting that all previous works (see Section 2 and Table 1) have only achieved unaccelerated estimates under various convexity assumptions. Moreover, the work [1] that appeared after our submit also achieves only unaccelerated algorithms. This fact demonstrates that it has not yet been possible to find such an accelerated method that is satisfied with only the direction of the gradient (without knowledge of the modulus, see the definition of the Order Oracle)... This includes among various methods with auxiliary low-dimensional minimization [2-4]. In all the accelerated methods known to us, the computation of the real gradient was assumed in one way or another. Therefore, the approach proposed in our paper is perhaps the only feasible scenario at the moment. Moreover, prior to our work, it was not clear that it was possible in any existing accelerated coordinate algorithm **to completely get rid of the knowledge of directional derivatives** and reduce everything to one-dimensional searches. It is worth noting that this observation is not trivial. For example, this was not the case with full-gradient accelerated algorithms. Moreover, we did not immediately succeed with coordinate algorithms, since there are already many of them. We had to find one for which it was possible to prove the corresponding statement. Further, we did not stop there and provided (see Appendix G) an algorithm that improves the already accelerated convergence estimates in the case of a low-dimensional problem. Finally, in Section 5 of our paper, we opened a new research direction (stochastic order oracle) by providing the first convergence results (of another algorithm), but already in the stochastic oracle concept.
>
> [1] Zhang, C., & Li, T. (2024). Comparisons Are All You Need for Optimizing Smooth Functions. arXiv preprint arXiv:2405.11454.
>
> [2] Drori, Y., & Taylor, A. B. (2020). Efficient first-order methods for convex minimization: a constructive approach. Mathematical Programming, 184(1), 183-220.
>
> [3] Narkiss, G., & Zibulevsky, M. (2005). SESOP: Sequential Subspace Optimization Method.
>
> [4] Hager, W. W., & Zhang, H. (2006). A survey of nonlinear conjugate gradient methods. Pacific journal of Optimization, 2(1), 35-58.
>
> With Respect,
>
> Authors

---

> > ### Comment · Reviewer_XqbR · 2024-08-10
> > **On the rebuttal**
> >
> > I thank the authors for their responses to my questions. I have read the responses carefully and I am inclined to retain my original scores.
> > And also, it must be interesting to see the algorithm in action real time, all the best to the startup!

---

> > > ### Author Response · Authors · 2024-08-12
> > >
> > > Dear **Reviewer XqbR**,
> > >
> > > We thank you for checking our response and for the positive rating.
> > >
> > > With Respect,
> > >
> > > Authors

---

### Author Rebuttal · Authors · 2024-08-04

Dear **Reviewers**, we thank you for taking the time to prepare reviews of our work. We have prepared detailed answers to the comments and questions that arose in the reviews. Our responses will be found under the official review. However, we would like to emphasize the highlights from your reviews.

- For example, **Reviewers XqbR** and **uSy5** advised to add experiments closer to reality. We would like to point out that our work was initiated due to a challenge faced by one of the co-authors in the realization of a startup (a smart coffee machine, see Appendix A1). Already in the process of preparing the paper, we started testing it. At the moment we have ready experiments on more than a real problem, where the coffee machine uses the algorithms presented in this paper, with a deterministic oracle. However, we have encountered some difficulties: since the coffee machine is not yet patented (we've started the patenting process), we have not been able to insert these experiments. Regardless, we thought we found a decent temporary alternative - **model examples that clearly verify the theoretical results**. Nevertheless, we realize that adding these results will further strengthen our work. Currently we expect the patenting process for the coffee machine to be completed in September-October this year, so in case of a positive decision (acceptance at the conference) on our paper, we are confident that we will have time to add the experiments on real problems to the camera-ready version.

- Whereas **Reviewer K16N** emphasized in the Summary section all the key points of our work that demonstrate the significance and novelty of the results. *We would like to describe them in more detail*: This work was motivated by a specific application problem (startup: smart coffee machine), but the presented results are not limited to it! The oracle concept under consideration has a number of applications, including perhaps the most popular setting, namely *Reinforcement learning with Human Feedback (RLHF)*. It is also worth noting that all previous works (see Section 2 and Table 1) have only achieved unaccelerated estimates under various convexity assumptions. This fact demonstrates that it has not yet been possible to find such an accelerated method that is satisfied with only the direction of the gradient (without knowledge of the modulus, see the definition of the Order Oracle)... This includes among various methods with auxiliary low-dimensional minimization [1-3]. In all the accelerated methods known to us, the computation of the real gradient was assumed in one way or another. Therefore, the approach proposed in our paper is perhaps the only feasible scenario at the moment. Moreover, prior to our work, it was not clear that it was possible in any existing accelerated coordinate algorithm **to completely get rid of the knowledge of directional derivatives** and reduce everything to one-dimensional searches. It is worth noting that this observation is not trivial. For example, this was not the case with full-gradient accelerated algorithms. Moreover, we did not immediately succeed with coordinate algorithms, since there are already many of them. We had to find one for which it was possible to prove the corresponding statement. Further, we did not stop there and provided already an algorithm that improves the already accelerated convergence estimates in the case of the low-dimensional problem (see Appendix G). Finally, in Section 5 of our paper, we opened a new research direction (stochastic order oracle) by providing the first convergence results of another algorithm, but already in the stochastic oracle concept.

- Finally, **Reviewers WKzP** and **ErqJ** drew attention to the proposed additional advantages of our approach. Namely, the advantage in the case of a high-dimensional problem, which is typical for the class of coordinate algorithms. Since we are based on the coordinate descent algorithm, we expectedly inherit this challenge. However, as you have already noticed below (in the Questions section), there is no explicit dependence on dimensionality in the final estimates (unlike previous work with Order Oracle). This fact is an advantage of our scheme (approach), since there are cases (e.g., *the asymmetric case: when the trace of the Hessian matrix has the same order as the largest eigenvalue*) where the dimensionality dependence is reduced (again, this is typical for the class of coordinate algorithms). In this case, our algorithm will be significantly superior to all previous algorithms, since in alternative algorithms the explicit dependence on dimension $d$ does not disappear anywhere.

We would like to thank **all Reviewers** once again for their interest in our work. We really appreciate it!

[1] Drori, Y., & Taylor, A. B. (2020). Efficient first-order methods for convex minimization: a constructive approach. Mathematical Programming, 184(1), 183-220.

[2] Narkiss, G., & Zibulevsky, M. (2005). SESOP: Sequential Subspace Optimization Method.

[3] Hager, W. W., & Zhang, H. (2006). A survey of nonlinear conjugate gradient methods. Pacific journal of Optimization, 2(1), 35-58.


Best regards,

Authors

---

### Decision · Program_Chairs · 2024-09-25

**Decision:**

Accept (poster)

**Comment:**

This paper formalizes an 'OrderOracle' model for function optimization where the function values or gradients are not available, and just comparison queries between function values at different points are permitted. The paper presents new optimization algorithms under this model and importantly, gives the first algorithm to achieve acceleration in this model.
The reviewers were split on how surprising the results in the paper are, but I think the model and the results (establishing that one could achieve essentially the convergence rates of randomized coordinate descent, without needed exact function values or gradients) are interesting, and I merit acceptance.